# Linear Transformers Implicitly Discover Unified Numerical Algorithms

**Patrick Lutz**[*]
Boston University
plutz@bu.edu

**Aditya Gangrade**[*]
Boston University
gangrade@bu.edu

**Hadi Daneshmand**[†]
University of Virginia
dhadi@virginia.edu

**Venkatesh Saligrama**
Boston University
srv@bu.edu

## Abstract

A transformer is merely a stack of learned datatodata maps—yet those maps can hide rich algorithms. We train a *linear, attention-only* transformer on millions of *masked-block* completion tasks: each prompt is a masked low-rank matrix whose missing block may be (i) a scalar prediction target or (ii) an unseen kernel slice for Nyström extrapolation. The model sees only input—output pairs and a mean-squared loss; it is given no normal equations, no handcrafted iterations, and no hint that the tasks are related. Surprisingly, after training, algebraic unrolling reveals the *same* parameter-free update rule across *all* three resource regimes (full visibility, bandwidth-limited heads, rank-limited attention). We prove that this rule achieves second-order convergence on full-batch problems, cuts distributed iteration complexity, and remains accurate with compute-limited attention. Thus, a transformer trained solely to patch missing blocks *implicitly discovers* a unified, resource-adaptive iterative solver spanning prediction, estimation, and Nyström extrapolationhighlighting a powerful capability of in-context learning.

## 1 Introduction

Models trained on next-token prediction achieve strong performance across various NLP tasks, including question answering, summarization, and translation (Radford et al., 2019). This multitask capability suggests an intriguing possibility: within structured mathematical contexts transformers might implicitly learn generalizable numerical algorithms solely through next-token prediction.

Next-token prediction can naturally be viewed as a form of matrix completioninferring missing entries by exploiting dependencies in observed data. Prior work extensively explores matrix completion under computational constraints, such as distributed data access Jordan et al. (2019), limited communication bandwidth Ma and Chen (2020), and low-rank recovery from partial observations Candès and Recht (2009); Candès and Tao (2010); Davenport and Romberg (2016). This raises a natural question: how do transformers implicitly handle such computational constraints to perform multitask learning?

*Can a neural network* invent *a numerical algorithm simply by learning to fill in missing data?*

Transformers excel at *in-context learning* (ICL), adapting to new tasks from a short prompt of examples Brown et al. (2020); Xie et al. (2022), and ICL serves as a simple subdomain to investigate the representational and learning properties of transformers. Recent studies suggest that for transformers trained on ICL tasks, the computations encoded by the weights resemble gradient-based methods Akyürek et al. (2023); Von Oswald et al. (2023); Ahn et al. (2023), typically in single-task scenarios. However, this analogy is limited: *gradient descent operates explicitly in parameter space, whereas transformers perform data-to-data transformations without direct access or updates to*

---

[*]These authors contributed equally to this work; code is available on github.com.

[†]Research performed under NSF Tripods Foundations of Data Science Institute Grant at Boston University.

*parameters*. Additionally, much of the exploratory focus in such work has been on the complexity of the regression task, while keeping the computational regime fixed. Thus, it remains unclear whether transformers implicitly develop distinct, unified numerical algorithms suited to diverse tasks and resource constraints.

**Masked-block Completion and Architectural Masks:** To probe this question, we train a linear transformer on masked-block completion tasks that hide one block of a low-rank matrix mirroring the classical Nyström completion problem. The transformer's task is to infer these missing blocks based solely on observable data and a mean-squared error objectivewithout explicit guidance about the underlying parameters or relationships between tasks. We impose three distinct architectural visibility constraints on transformers, each reflecting practical computational limitations common in large-scale optimization: *centralized* (full visibility), *distributed* (restricted communication), and *computation-limited* (restricted complexity via low-dimensional attention). Each regime employs the same underlying transformer architecture, differing only minimally in their attention masks.

**Emergence of a Unified Algorithm:** Remarkably, despite training independently under these distinct computational constraints, we find that transformers implicitly uncover the same concise, two-line iterative update rule. This unified algorithm, termed EAGLE, emerges consistently across constraints, exhibiting strong theoretical and empirical properties: it achieves second-order convergence on full-batch problems, matches classical methods in centralized settings, significantly reduces communication complexity in distributed settings and remains accurate under compute constraints.

**Summary of Contributions.** The main contributions of this work are:
• *Unified masked-block completion benchmark:* A single training framework integrating multiple prediction or inference tasks into a unified interface.
• *Resource-aware transformer architectures:* Transformer-based models naturally adapt to centralized, distributed, and computation-limited environments through simple architectural constraints.
• *Implicit numerical algorithm discovery:* Transformers implicitly discover a unified, efficient numerical solver, EAGLE, that achieves second-order convergence on full-batch problems and consistently matches or outperforms classical methods across completion, extrapolation, and estimation tasks under varying resource constraints. Our theoretical results uniformly apply across all these tasks, highlighting the broad applicability of the discovered algorithm.

These findings position transformers trained on block completion as powerful tools for uncovering numerical algorithms, offering a promising route toward developing adaptive, data-driven solvers.

## 2 Related Work

We focus on literature most pertinent to viewing transformers as fixed data-to-data transforms whose forward pass exposes emergent algorithms.

*Implicit algorithm learning.* Transformers have been shown to recover $1^{st}$- and $2^{nd}$-order methods for least squares, dual GD for optimal transport, and TD updates for RL (Von Oswald et al., 2023; Ahn et al., 2023; Daneshmand, 2026; Wang et al., 2025; Fu et al., 2024; Giannou et al., 2023). Weight-level circuit studies reverse-engineer copy-and-addition induction heads (Elhage et al., 2021; Olsson et al., 2022), but are still confined to token-manipulation algorithms. RNNs and MLPs display related behavior (Siegelmann and Sontag, 1992; Tong and Pehlevan, 2024), yet attention yields depth-aligned, easily inspected updates (Garg et al., 2022). Most prior work is therefore limited to first-order rules and a single, centralized computational regime.

*Task vs. regime.* Earlier in-context studies (Akyürek et al., 2023; Bai et al., 2023; Min et al., 2021) vary the regression tasksampling while the hardware regime stays fixed. We take the opposite view: we *fix* one low-rank matrix-completion task and show that the same transformer discovers a rule that adapts automatically as compute, memory, or communication budgets are tightened.

*Data-space vs. parameter-space.* Prior analyses interpret the forward pass of a transformer trained on an ICL task as updating an underlying hidden *parameter* for some model of the training data relationships. However, in reality, a transformer is a *data-to-data map*, with no explicit supervision of underlying model of parameters, which is the viewpoint we adopt. Weight-level circuit studies (Elhage et al., 2021; Olsson et al., 2022) of trained transformers also take this view, but are focused on tasks like copying or addition; we instead are focused on numerical linear algebra tasks in varying computational settings.

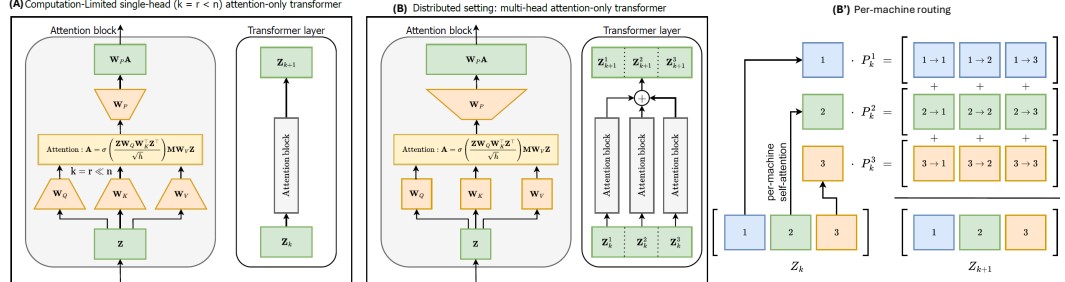

Figure 1: **Architectural regimes studied in this work.** **(A)** *Computation-limited*: a single-head attention-only transformer whose query, key , value & projections are restricted to a low-rank embedding of dimension $k = r \ll n$, i.e., an explicit bottleneck that reduces the per-layer cost from $\Theta(n^2 d)$ to $\Theta(n\, r\, d)$; *Unconstrained*. The embedding dimension is set to $k = n + n'$. **(B)** *Distributed*: a multi-head transformer in which each head operates on data stored on a separate machine; the heads run local attention and their outputs are aggregated. **(B')** *Per-machine routing*: detailed view of the distributed setting showing how each machine $\mu$ forms its local projection $P_k^\mu$ and contributes to the next-layer representation $Z_{k+1}$.

*In-context regression limits.* Theoretical lenses that view in-context learning as linear regression (Akyürek et al., 2023; Bai et al., 2023) recover gradient-descent dynamics with $\sqrt{\kappa}$ dependence on the condition number. Vladymyrov et al. (2024) show that, with suitable parametrization, preconditioning yields second-order dynamics and $\log \kappa$ convergence. We show that transformer training naturally converges to such parametrizations in practice, with this behavior robust to compute and memory constraints. Theoretically, we prove that the recovered EAGLE method enjoys favorable guarantees across all studied settings, and further validate these empirically.

*Links to classical numerical algorithms.* Structurally, the recovered EAGLE iterations bear intimate relationship to the Newton-Schulz method for inverting positive matrices (Higham, 1997), but extend this to solve non-positive linear systems. In the centralised and sketched settings, this underlying relationship lends EAGLE a second-order convergence rate, with iteration complexity depending only logarithmically on the condition number. This is particularly interesting since usual implementations of Nyström approximations for kernel methods all focus on Krylov-based solvers (Williams and Seeger, 2001; Halko et al., 2011; Gittens and Mahoney, 2016), and EAGLE may lend new approaches to the same.

*Distributed and sketched solvers.* Block-CG, communication-avoiding Krylov (Demmel et al., 2013; Hoemmen, 2010) and sketch-and-solve techniques (Clarkson and Woodruff, 2017; Woodruff, 2014) dominate practice but remain first-order, and have net communication requirements scaling with the (joint) covariance of the data. We characterise the distributed EAGLE performance in terms of a 'data diversity index' $\alpha$, and find significantly smaller communication (and iteration) complexity when $\alpha^{-1}$ is much smaller that this joint condition number.

## 3 Method

We explain how data are generated and encoded, how architectural constraints enforce three resource regimes, and how we extract an explicit numerical solver from trained weights.

**Block Completion Setup.** We generate a low-rank matrix task by first drawing a matrix of the form

$$X = \begin{bmatrix} A & C \\ B & D \end{bmatrix} \in \mathbb{R}^{(d+d')\times(n+n')}, \quad \operatorname{rank}(X) = \operatorname{rank}(A), \tag{1}$$

and then feeding the transformer a prompt matrix $Z_0$ in which the lower-right grey block $D$ is masked, i.e., set to zero. Given the visible blocks $A, B, C$, the model must reconstruct $D$ with low $\ell_2$ error. This single prompt template generalizes Nyström extrapolation and scalar regression (Appx. B). We train on both exact samples and noisy variants obtained by adding Gaussian noise with variance $\sigma^2 \in \{0, 10^{-2}\}$, so the low-rank assumption is approximate but realistic. Note that the Nyström approximation is the minimum-rank completion that matches observed entries, even if $X$ is full rank.

**Data generation.** We construct rank-$s$ matrices $X \in \mathbb{R}^{(d+d')\times(n+n')}$ by sampling $R_1 \in \mathbb{R}^{(d+d')\times s}$ and $R_2 \in \mathbb{R}^{(n+n')\times s}$, each with rows drawn i.i.d. from $\mathcal{N}(0, \Sigma)$, and setting $X = R_1 R_2^\top / \sqrt{s}$. To

control the difficulty of the problem, we choose $\Sigma$ to be diagonal with entries $\Sigma_{ii} = \alpha^i$, where $\alpha < 1$, inducing strong anisotropy. We use the following parameters throughout: $n = d = 18$, $n' = d' = 2$, $s = 10$ and $\alpha = 0.7$. When applicable, we add Gaussian noise of variance $0.01$ to $X$.

**Linear-attention transformer.** We employ the linear-attention variant of transformers (Ahn et al., 2023; Von Oswald et al., 2023; Wang et al., 2025). Each layer $\ell$ applies multi-head attention with a residual skip connection, to transform $Z_0 = \begin{bmatrix} A & C \\ B & 0 \end{bmatrix}$ to $Z_\ell := \begin{bmatrix} A_\ell & C_\ell \\ B_\ell & D_\ell \end{bmatrix}$ via the iterative structure

$$Z_{\ell+1} = Z_\ell + \sum_{h \in [1:H]} \text{Attn}_\ell^h(Z_\ell), \quad \text{Attn}_\ell^h(Z) = \left(ZW_{Q,\ell}^h(ZW_{K,\ell}^h)^\top \odot M_\ell^h\right) Z W_{V,\ell}^h W_{P,\ell}^{h\top}, \quad (1)$$

where $W_Q, W_K, W_V, W_P \in \mathbb{R}^{n \times k}$ are query, key, value, and projection matrices, and the fixed mask $M_\ell^h = \begin{bmatrix} \mathbf{1}_{(d+d')}\mathbf{1}_d^\top & 0_{(d+d') \times d'} \end{bmatrix}$ blocks the flow of information from the incomplete $(C_\ell, D_\ell)$ column in $Z_\ell$ towards the visible column. Models are trained to minimize mean-squared error on $D$; training details are in §B.2. A schematic of the architecture appears in Fig. 1(A).

**Computational Regimes via Architectural Constraints.** We study three distinct computational regimesunconstrained (or centralized), computation-limited (low-dimensional attention), and distributed (restricted memory per machine)each encoded into the architecture through explicit attention and dimensionality constraints. Figure 1 summarizes these architectural regimes.

*Unconstrained.* No visibility or dimension constraints are imposed; each token attends to all others. We set the embedding dimension $k = n + n'$.

*Computation-Limited.* To emulate memory- or latency-bounded hardware we enforce a *low-rank attention* constraint: the query and key matrices have embedding size $k = r \ll n$, and the value and projection size $r + n'$ (we use $r = 5 \approx n/4$; see Fig. 1A). This bottleneck compresses the input and cuts the per-layer cost of recovered algorithms from $\Theta(nd^2)$ to $\Theta(ndr)$. (See §B.3 for details).

*Distributed Computation.* The prompt $Z_0$ distributes $X$ across $M$ machines through a block structure

$$Z_0 = \begin{bmatrix} \cdots & X^\mu & X^{\mu+1} & \cdots \end{bmatrix} \in \mathbb{R}^{(d+d') \times M(n+n')}, \; where \; X^\mu = \begin{bmatrix} A^\mu & C \\ B^\mu & D \end{bmatrix}, \mu \in [1:M]$$

Each head is assigned to a machine $\mu$, and all but the $\mu$th block in its query, key matrices are zeroed, enforcing local self-attention. Each $\mu$ then 'transmits' information to others through the value, projection matrices, and incoming transmissions are summed to generate $Z_{\ell+1}$. The full algebraic form, matrix partitioning and communication setups are provided in §B.4.

**Algorithm extraction.** We generate explicit algorithms from a trained transformer by progressively simplifying its weights until a concrete update rule emerges. The steps taken are (also see §B.5):
• *Weight quantization.* Cluster coefficients and sparsify by dropping the values below $\leq \tau$. We then evaluate the performance with simplified weights to confirm no loss in performance.
• *Matrix Property Tests.* Check whether resulting weight matrices are random, sparse, or low-rank.
• *Scaling Laws.* Identify how the transformer scales weight matrices layer-by-layer.

```
1: function UPDATE(A, B, C, D)                1: Input: {A_0^μ, B_0^μ}_{μ=1}^M, query/output (C_0, D_0)
2:   Ã ← AS,   B̃ ← BS                         2: for ℓ = 0, ..., L − 1 do
3:   ρ ← ‖Ã‖_2^{-2}                            3:     C_sum ← 0, D_sum ← 0
4:   A' ← A − ηρ ÃÃ^⊤ ÃS^⊤                     4:     for μ = 1, ..., M in parallel do
5:   B' ← B − ηρ B̃Ã^⊤ ÃS^⊤                    5:         (A_{ℓ+1}^μ, B_{ℓ+1}^μ, C'^μ, D'^μ) ← UPDATE(A_ℓ^μ, B_ℓ^μ, C_ℓ, D_ℓ)
6:   C' ← C − γρ ÃÃ^⊤ C                        6:     end for
7:   D' ← D + γρ B̃Ã^⊤ C                       7:     C_{ℓ+1} ← (1/M) Σ_μ C'^μ, D_{ℓ+1} ← (1/M) Σ_μ D'^μ
8:   return (A', B', C', D')                   8: end for
9: end function                               9: Return D_L
```

**(A) Update: Identical for all regimes**    **(B) Fusion: Lightweight Communications**

Algorithm 1: **Emergent Algorithm for Global Low-rank Estimation (EAGLE).** *Left:* the numerical kernel UPDATE runs unchanged on every machine. *Right:* the outer loop calls UPDATE, and in the distributed regime ($M > 1$), it averages the resulting query updates ($C'$) and outputs ($D'$). $M = 1$ in the unconstrained and computation-limited settings. In the former, $S = I_n$, while in the latter, $S \in \mathbb{R}^{(d+d') \times r}$ is composed of random orthogonal rows. The values $\eta, \gamma$ are global constants (see below for their values).

In the unconstrained and compute-limited settings, an architecture with one head per layer suffices to achieve competitive performance, while in the distributed setting, we use one head per machine. After these simplifications every regime collapses to the *same* two-line, parameter-free update rule (Eq. 2 in § 4), providing a direct algorithmic interpretation of the transformer's in-context computation.

| layer | U×$10^4$ | D×$10^4$ | C-L×$10^4$ |
|---|---|---|---|
| 0 | 0.00 | 0.00 | 0.00 |
| 1 | 2.36 | 0.62 | 0.01 |
| 2 | 5.27 | 1.40 | 0.02 |
| 3 | 3.70 | 1.28 | 0.13 |
| 4 | 2.16 | 1.32 | 0.14 |

Table 1: Differences across iterations between transformer and extracted algorithm (squared Frobenius norm divided by size of $Z$) in the unconstrained (U), distributed (D), and compute-limited (C-L) settings. Mean *times* $10^4$ across 10 seeds is reported.

## 4 Emergent Algorithm

**One method, three resource regimes.** Across all three architectural constraints, the trained 4-layer transformer achieves a 1050 times reduction in test MSE (Fig. 3 a). Moreover, algebraic interpretation of the learned weights shows it implements the same two-line scaling transformation across constraints. Algorithm 1 visualises the result, which we call EAGLE (Emergent Algorithm for Global Low-rank Estimation). The blue UPDATE box is identical in all scenarios and across all machines, while the outer grey look captures the information fusion in the distributed setting (i.e., if $M > 1$). The matrix $S$ in UPDATE is an orthogonal *sketching matrix* (see below). Table 3 shows that this extracted algorithm reproduces the exact layer-wise activations of the transformer to within $6 \cdot 10^{-4}$ error, demonstrating its fidelity. We now discuss how this algorithm is derived in the three settings, focusing on the noiseless case. However, our findings remain valid under modest data noise ($\sigma^2 = 0.01$, see §C.2). Moreover, §D shows how Algorithm 1 can be adopted for parameter estimation.

**Unconstrained setting.** Because token updates are not restricted, the weight patterns are the cleanest to interpret here, and will re-appear in the distributed and compute-limited regimes.

• *Emergent weight structure.* During the extraction procedure we progressively pruned attention heads and quantised weights while tracking validation loss. Remarkably, *one head per layer* is already sufficient: pruning from eight to a single head has little effect on test MSE. After pruning, the weight products $W_{Q,\ell}W_{K,\ell}^\top$ and $W_{V,\ell}W_{P,\ell}^\top$ collapse to (almost) diagonal matrices.

$$W_{QK,\ell} := W_{Q,\ell}W_{K,\ell}^\top \approx \operatorname{diag}(\alpha_\ell^1 I_n, \, 0_{n'}), \qquad W_{VP,\ell} := W_{V,\ell}W_{P,\ell}^\top \approx \operatorname{diag}(\alpha_\ell^2 I_n, \, \alpha_\ell^3 I_{n'}),$$

where only the three scalars $\alpha_\ell^{1,2,3}$ vary from layer to layer. Figure 2 (left) visualises a typical pair; layer-wise statistics across 10 seeds are reported in §C.3. This near-block-diagonal structure lets us algebraically reduce the transformers forward step to the update shown in Fig. 1.

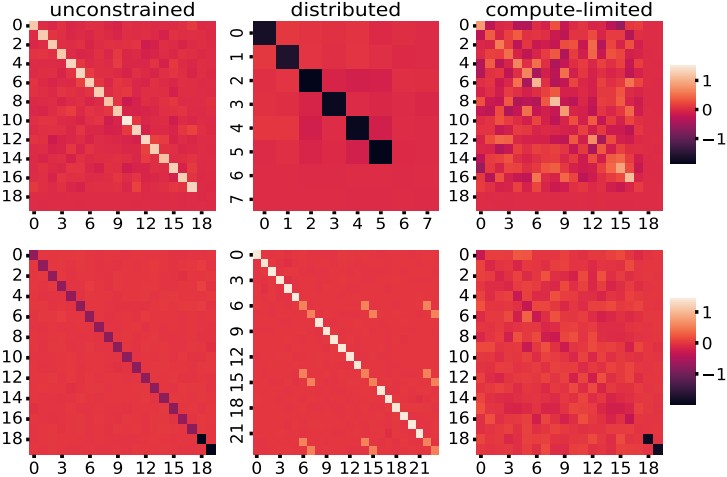

Figure 2: Block structure of $W_{QK,\ell}$ (*top*) and $W_{VP,\ell}$ (*bottom*) learned in the three regimes. Example shown for $(n, d, d', n') = (18, 18, 2, 2)$; distributed run uses $M = 3$ workers with per-worker $n = 6$, and all of the $\{W_{VP}^\mu\}_{\mu=1,2,3}$ are collated together with each of the three block-wise rows corresponding to one head, while omitting all (null) non-$\mu$ blocks in $W_{QK}^\mu$. Off-diagonal blocks appear *only* in $W_{VP,\ell}^\mu$s, and are identical suggesting structure of messaging. Statistics over 10 seeds are in §C.3.

- *From weights to* UPDATE. The additive steps in UPDATE are due to the residual connection. Further, the diagonal forms of $W_{QK,\ell}$ and $W_{VP,\ell}$ imply that (see §C.4), the attention block acts as

$$\text{Attn}_\ell(Z_\ell) = \begin{bmatrix} \alpha_\ell^1\alpha_\ell^2 A_\ell A_\ell^\top A_\ell & \alpha_\ell^1\alpha_\ell^3 A_\ell A_\ell^\top C_\ell \\ \alpha_\ell^1\alpha_\ell^2 B_\ell A_\ell^\top A_\ell & \alpha_\ell^1\alpha_\ell^3 B_\ell A_\ell^\top C_\ell \end{bmatrix}. \tag{2}$$

The form of the iterative update in Alg. 1 with $M = 1, S = I_n$ is then immediate.

- *Rationale for Weights.* Further, the values $\alpha_\ell^{1,2,3}$ see two persistent structural effects: throughout, $\alpha_\ell^1\alpha_\ell^2 \approx \eta\|A_\ell\|_2^{-2}$, and $\alpha_\ell^1\alpha_\ell^3 \approx \gamma\|A_\ell\|_2^{-2}$, where $\|A_\ell\|_2$ is the spectral norm of $A_\ell$ (i.e., largest singular value, tuned to the largest such value typically seen in a training batch), and $\gamma, \eta$ are the *constants* $\eta \approx 1$ and $\gamma \approx 1.9$. Thus, the scale of these weights is determined by $\|A_\ell\|^{-2}$, which is used in a fixed way by the method. Figure 11 shows that $\alpha_\ell^1\alpha_\ell^2\|A_\ell\|^2$ is constant across layers.

- *Understanding the method.* Note that the transformation $A \mapsto (I - \rho\eta AA^\top)A$ contracts large singular values of $A$ more than small singular values. The net effect of this repeated action on $A_\ell$ is that for large $\ell$, all initially nonzero singular values of $A_\ell$ converge to one another, i.e., the matrix $A$ becomes well-conditioned (see Fig. 3). This aspect of the method is reminiscent of the Newton-Schulz method for matrix inversion (Schulz, 1933; Ben-Israel, 1965; Higham, 2008; Fu et al., 2024). The overall structure of the iterations can then be seen as a 'continuous conditioning update' for $A_\ell$. The iterations for $C_\ell, D_\ell$ are reminiscent of gradient descent, adapted to the varying $A_\ell$.

- *Relation to prior work.* Von Oswald et al. (2023) dubbed the same update GD$^{++}$ in the scalar case ($d' = n' = 1$) and interpreted it as pre-conditioned gradient descent. In our opinion, there are two deficiencies in this viewing of the method. Firstly, gradient based methods implicitly assume that one is optimising a parameter (i.e., searching for a $W$ such that $B \approx WA$, and imputing $D \approx WC$), whereas the transformer has not been supervised in a manner that reveals the existence of an underlying parameter. In other words, a gradient descent type interpretation is not a behavioural (in the sense of Willems (1991)) description of the recovered method. Secondly, our analysis shows that unlike gradient descent, the recovered method is closer to a *NewtonSchulz conditioning loop* wrapped around direct prediction. For these reasons, we refer to it as the EAGLE method in the rest of the paper. §**??** contrasts the two viewpoints in detail.

**Distributed setting.** Following the unconstrained setting, we train with a single head per machine.

- *Communication Structure* As detailed in §3, for each head, the block structure of the $(n + n') \times M(n + n')$ value-projection matrix $W_{VP,\ell}^\mu$ governs what machine $\mu$ *sends* to other machines. Fig. 2 (middle) reveals two striking regularities in this matrix.

    – *Within-machine blocks.* Every diagonal block equals $\text{diag}(\alpha_\ell^1 I_n, \ \alpha_\ell^2 I_{n'})$ exactly the same form as in the unconstrained model, with the *same* pair $(\alpha_\ell^1, \alpha_\ell^2)$ across all machines.

    – *Across-machine blocks.* Off-diagonal blocks are $\text{diag}(0_n, \ \alpha_\ell^2 I_{n'})$: i.e. only the incomplete columns $(C_\ell, D_\ell)$ are transmitted, and with the very same factor $\alpha_\ell^2$ as the within machine blocks.

- *Resulting algorithm.* Since $W_{QK,\ell}^\mu$ and the local block of $W_{VP,\ell}^\mu$ have the same structure as the unconstrained setting, each machine executes the same local iteration as in the UPDATE method. Further, via the off-diagonal blocks in $W_{VP,\ell}^\mu$, each machine transmits only the $O(n'(d + d'))$ entries of its update to $C_\ell, D_\ell$, captured in the $C', D'$ outputs in Alg. 1. Since attention heads are simply added, every machine averages its received blocks, leading to $C_{\ell+1} = M^{-1}\sum_\mu C'^\mu, \ D_{\ell+1} = M^{-1}\sum_\mu D'^\mu$. The shared columns are thus identical across all $\mu$ at all $\ell$, recovering Alg. 1 with $S = I_n$ and $M > 1$.

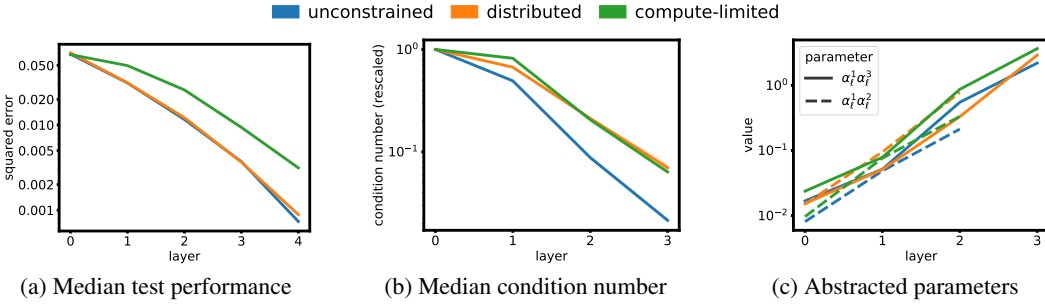

(a) Median test performance      (b) Median condition number      (c) Abstracted parameters

Figure 3: The trained transformer solves matrix completion with a unified algorithm over all three computational settings. The evolution of key quantities throughout the transformer layers illustrate the remarkable similarity between the latent algorithms. Mean across 10 training seeds is reported.

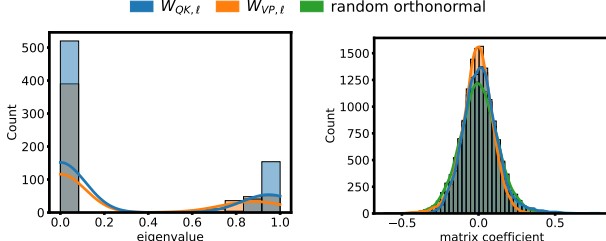
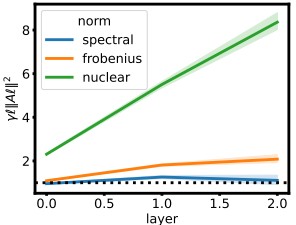

Figure 4: The computation-limited transformer implements pseudo-random sketching. The figure matches the candidate sketch matrices across all layers against common sketch characteristics (randomness, clustered eigenvalues). Left: Distribution of eigenvalues. Right: Distribution of coefficients.

Figure 5: The transformer learns to normalize the batch-maximum spectral norm of $A_\ell$ observed during training. The plot reports $\alpha_\ell^1 \alpha_\ell^2 \max_{b \in [B]} \|A_\ell^{(b)}\|^2$, where the maximum is taken over each batch.

- *Communication cost.* Under the star topology in Alg. 1, each machine communicates $O\big((d+d')\,n'\big)$ floats per round; for scalar prediction $(d' = n' = 1)$ this is the advertised $2d$-float message. Notably, the transformer recovers this sparse pattern *without* any explicit communication constraint. Exploring how stronger topological or bandwidth limits shape the learned algorithm is an open problem.

**Computation-limited setting.** Here the query, key, value and projection matrices are rank-constrained: $W_{Q,\ell}, W_{K,\ell} \in \mathbb{R}^{(d+d') \times r}$ and $W_{V,\ell}, W_{P,\ell} \in \mathbb{R}^{(d+d') \times (r+n')}$ with $r \ll n$. Similar low-rank constraints commonly arise when attention head dimensions are smaller than the transformer's overall embedding size.

- *Emergent weight structure.* Again, only one head per layer is needed. The $(n + n') \times (n + n')$ matrices $W_{QK,\ell}$ and $W_{VP,\ell}$ share three universal properties (Fig. 2)
    - *Block-diagonal form.* Both are block-diagonal; the cross blocks $(n \times n')$ and $(n' \times n)$ vanish as in the unconstrained regime. The $(n' \times n')$ block of $W_{VP,\ell}$ is a scaled identity, exactly as before.
    - *Rank-$r$ top left block.* The leading $n \times n$ block of each matrix has numerical rank $r$ $(< n)$ and the two blocks are similar up to a sign when rescaled to spectral norm 1 (relative difference: 0.28).
    - *Random sketch spectrum.* The eigenvalues and entry distribution of the top-left block match those of $SS^\top$ where $S$ is a random $n \times r$ orthogonal matrix (Fig. 4).
- *Sketching interpretation.* The transformer therefore materializes an *orthogonal row sketch* $S \in \mathbb{R}^{n \times r}$ within the 'top left' blocks of its $W_{QK}$ and $W_{VP}$ matrices. This sketch acts upon the columns within the $(A_\ell, B_\ell)$ blocks of the state $Z_\ell$, and the output of the attention block is structured as

$$\text{Attn}_\ell(Z) = \begin{bmatrix} \alpha^1 (AS)(AS)^\top (AS) S^\top & \alpha^2 (AS)(AS)^\top C \\ \alpha^1 (BS)(AS)^\top (AS) S^\top & \alpha^2 (BS)(AS)^\top C \end{bmatrix}, \tag{3}$$

where $\alpha^1, \alpha^2$ are constants depending on $\|A\|^{-2}$. In other words, the update first computes the sketches $AS, BS \in \mathbb{R}^{d \times r}$ of the 'complete' columns, and then proceeds with the update as in the unconstrained case, lifting them back to $n \times d$ by the terminal $S^\top$. Setting $M = 1$ and $S = S_\ell$ in Algorithm 1 recovers the exact layer dynamics.

- *What the sketch buys.* Under the sketch, $\tilde{A}_\ell = AS \in \mathbb{R}^{d \times r}$ is only $r$ columns wide, reducing the per-layer flop count from $O(n^2 d)$ to $O(nrd)$. Although this sketched update causes the iteration complexity of the method to increase, the overall hope is that the lower per-iteration cost may give a better total runtime. This aligns with the original motivation in Vaswani et al. (2017), where low-rank constraints on $W_{Q,\ell}, W_{K,\ell}, W_{V,\ell}$, and $W_{P,\ell}$ were explicitly introduced for computational efficiency.

# 5 Evaluation of EAGLE

Having identified the EAGLE update underlying the transformer weights, we move on to studying how well the extracted algorithm performs in the three regimes, both theoretically and empirically. All ablations are deferred to §E, while theoretical proofs appear in §F. Unless stated otherwise, all numerical evaluations in this section use data sizes $n = d = 240$ and $n' = d' = 2$, condition number $\kappa(A) = 10^2$, $\text{rank}(A) = 240$, average performance across 50 runs is reported.

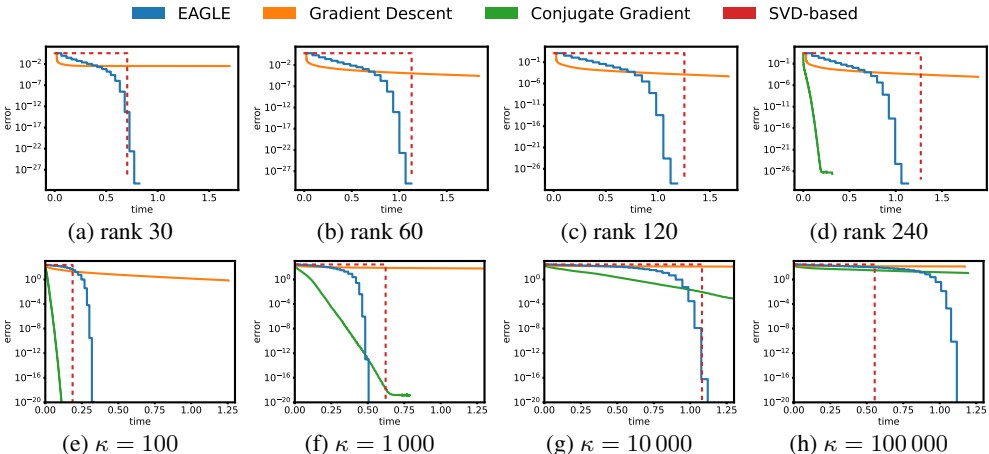

Figure 6: EAGLE shows second order convergence in the unconstrained regime. It extends to low-rank completion and scales logarithmically with the condition number $\kappa$.

## 5.1 Unconstrained (centralized) setting

**Second-order convergence guarantee.** We discuss the theoretical convergence properties of EAGLE in the centralized case. Let $\overline{\sigma}(M)$ and $\underline{\sigma}(M)$ denote the largest and smallest positive singular values of a matrix $M$ and let $\kappa(M) = \overline{\sigma}(M)/\underline{\sigma}(M)$ denote the effective condition number.

**Theorem 1.** *For any $X$, let $\hat{D}_*$ be the Nyström estimate for $D$ and $\kappa = \kappa(A_0)$. If $\eta = 1/3, \gamma = 1$, then under EAGLE with $S = I_n, M = 1$, for any $\varepsilon > 0$, there exists*

$$L = O\big(\log \kappa + \log \log(\varepsilon^{-1}\sqrt{d'}\|W_*\|_F\|C\|_F)\big)$$

*such that $\forall \ell \geq L, \|D_L - \hat{D}_\star\|_F \leq \varepsilon$, where $W_* = BA(AA^\top)^\dagger$ is the Nyström parameter (§B.1).*

*Mechanism.* We show Thm. 1 in §F.1 by arguing that each iteration shrinks $\kappa_\ell := \kappa(A_\ell)$ by at least a constant factor, and that once $\kappa_\ell \leq 2$, then $\kappa_\ell - 1$ decays supergeometrically. The error is controlled by developing a telescoping series with terms decaying with $(\kappa_\ell - 1)$. The $\log \log(\varepsilon^{-1})$ dependence in $L$, i.e., the quadratic rate, is related to the quadratic decay in $\kappa_\ell - 1$, which in turn arises since the EAGLE iterations for $A_\ell$ bear a strong relationship to the classical Newton-Schulz method (Higham, 2008, Ch. 5,7). We note that a simple $\|D_\ell - D_{\ell-1}\|_F < \tau$ stopping rule suffices in practice.

*Positioning vs. classics.* Direct inversion via QR-decomposition or SVD costs $O(\min n^2 d, d^2 n)$ once, independent of $\kappa$. Krylov methods such as the Conjugate Gradient method (CG, Hestenes and Stiefel, 1952) and gradient descent (GD) need $\kappa \log(\varepsilon^{-1})$ and $\kappa^2 \log(\varepsilon^{-1})$ iterations and $O(nd)$ time per-iteration to compute matrixvector products. By contrast, EAGLE achieves quadratic convergence with iteration counts scaling only with $\log(\kappa)$, albeit with $\min(n^2 d, d^2 n)$ cost per iteration.

**Empirical protocol.** To evaluate the empirical performance of the recovered algorithm, we benchmark EAGLE against a SVD-based solver (`torch.linalg.lstsq`), the Conjugate Gradient method and gradient descent (GD) on synthetic $A \in \mathbb{R}^{n \times n}$ (see Appx. E.1 for details). In particular, we study

1. *Error vs. wall-clock time.* Visually confirms the second-order convergence predicted by Theorem 1.
2. *$\kappa$-sweep.* Time to reach $\varepsilon = 10^{-20}$ for $\kappa \in \{10^2, \ldots, 10^5\}$; visualises the $\log \kappa$ vs. $\sqrt{\kappa}$ gap.
3. *Rank-deficient check.* Time to reach $\epsilon = 10^{-20}$ highlights robustness to rank-deficiency.

Figure 6 presents (1)(3); further experiments including extended size sweeps are in §E.3.

**Take-away.** The transformer-extracted update converges *quadratically* with only a $\log \kappa$ penalty— $\sim 100\times$ fewer iterations than CG at $\kappa = 10^4$ while retaining $O(\min(nd^2, dn^2))$per-iteration cost. This positions EAGLE in a previously unexplored region of the speedaccuracy trade-off: an iterative solver that matches SVD-based performance up to a $\log$ factor.

## 5.2 Distributed setting

**Diversity drives the rate.** Besides the local condition numbers $\kappa^\mu = \kappa(A^\mu)$, convergence depends on how *distinct* the column spaces on different workers are. We capture this via:

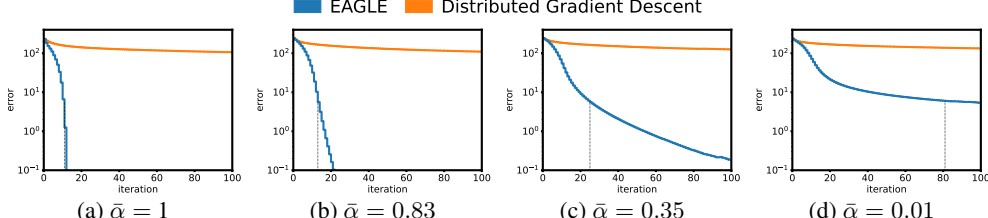

(a) $\bar{\alpha} = 1$  (b) $\bar{\alpha} = 0.83$  (c) $\bar{\alpha} = 0.35$  (d) $\bar{\alpha} = 0.01$

Figure 7: In the distributed setting, EAGLE shows significant improvements in iteration count. The dotted line indicates the iteration where the error reaches 6; as the theory suggest, the convergence depends linearly on $\alpha$.

**Definition 1.** *(Diversity index) Let $P^\mu$ be the orthogonal projection onto the range of $A^\mu$. The diversity index $\alpha$ of $\{A^\mu\}_{\mu=1}^M$ is the smallest positive eigenvalue of the average projection $\frac{1}{M}\sum_{\mu=1}^M P^\mu$.*

The diversity index $\alpha$ is large when the projectors cover their joint range uniformly. Conversely, $\alpha$ is small when the ranges are highly overlapping yet enlarge the joint span, leaving "innovation" directions weakly covered. The diversity index $\alpha$ captures the convergence scale of distributed EAGLE via the following result shown in §F.2.

**Theorem 2.** *Let $\kappa_{\max} = \max_\mu \kappa^\mu$. In the noiseless case, i.e., when $\exists W_*$ such that $[B \quad D] = W_* [A \quad C]$, then under EAGLE with $\eta = 1/3, \gamma = 1, S = I_n, M \in \mathbb{N}$, for any $\varepsilon > 0$, there exists*

$$L = O(\log(\kappa_{\max} + \alpha^{-1} \log(\sqrt{d'}\|C\|_F \|W_*\|_F/\varepsilon)))$$

*such that $\forall \ell \geq L, \|D_L - D\|_F \leq \varepsilon$.*

*Mechanism.* After $\log \kappa_{\max}$ iterations, every local $A_\ell^\mu$ has near-unit singular spectrum. Subsequent progress is controlled by the condition number of the *average energy matrix* $\bar{E}_\ell = M^{-1} \sum_\mu A_\ell^\mu A_\ell^{\mu\top}$, which tends to $\alpha^{-1}$. Second-order convergence re-emerges in practice when $\alpha = 1$. Let us note that $\alpha^{-1}$ is always smaller than $\kappa(\bar{E}_0)$ and may be much smaller than the same.

**Baselines.** Our main interest is in comparing with distributed baselines that operate with $O(d)$ units of communication per round—any more, and we may simply share the data across all machines into one central server. For this reason, QR/SVD decompositions and the Conjugate Gradient method are unavailable, which leaves gradient descent (GD) as the main competitor. GD and EAGLE have identical communication costs, transmitting $O((d + d')n')$ floats per iteration. However, the iteration complexity of GD scales with $\kappa(\bar{E}_0)$, which is always greater than $\alpha^{-1}$, and may be much larger. As a result, EAGLE has a strong advantage in practical strongly communication-limited scenarios.

**Empirical protocol.** We vary the two rate-determining parameters $M$ and $\alpha$, and report error against iteration in Figs. 8,7:

1. *$M$-sweep.* $M = 1, 3, 5, 8$ workers among which a fixed set of data is split ($d = n = 1000$, $\kappa_{\max} = 10^3$, $\alpha \approx 1$).

2. *$\alpha$-sweep.* Four synthetic datasets, $\kappa_{\max} = 10^3$, $M = 3$; cross-machine left-subspace alignment is tuned yielding varying $\alpha$ centered around 4 means $\bar{\alpha} \in \{1, 0.83, 0.35, 0.01\}$.

**Take-away.** Distributed EAGLE generally converges at first order and its iteration count grows linearly with $\alpha^{-1}$ (Fig. 7). Further, for controlled $\alpha$, the resulting iteration complexity is independent of the number of workers $M$ (Fig. 8). This confirms our theoretical result. In contrast, gradient based methods require $10100\times$ more rounds. Experiments on best-of-both hybrid (EAGLE until $A$ stabilizes, then GD) are detailed in §E.2.

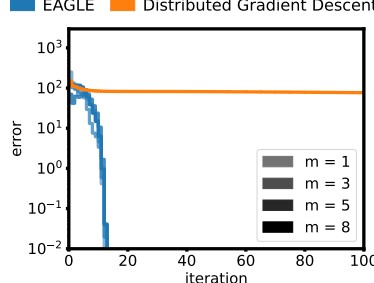

Figure 8: Iteration complexity of distributed EAGLE is independent of the number of workers $M$.

### 5.3 Computation-limited sketching

**Sketch-aware iterations.** We now turn to iterations driven by the sketched columns $\tilde{A}_\ell := S_\ell^\top A$, $\tilde{B}_\ell := S_\ell^\top B$ via the i.i.d. orthogonal sketches $S_\ell \in \mathbb{R}^{n \times r}$ ($r \ll n$). The isotropicity

of the sketch ensures that the singular-spectrum of $A$ is conditioned in roughly the same as the centralized setting (§5.1), up to a slowdown of a $\approx r/n$ factor.

**Baselines & Protocol.** The standard competitor is Stochastic Gradient Descent (SGD): GD run on an $r$-column sketch. It shares the $n/r$ spectral slowdown but keeps its native $\kappa^2$ dependence. For efficiency, our implementation of the resource-constraint EAGLE uses random row sampling matrices as $S$. We vary the rank, the single rate-controlling parameter $r$:

1. *$r$-rank sweep.* We vary the rank $r \in \{n/8, n/4, n/2, n\}$ in Figure 18.
2. *Per-iteration cost.* The per-iteration wall-clock time decreases with sketch size: 21, 16, 7 and 5 milliseconds for $s = 240, 120, 60$ and 30, respectively.

**Take-away.** Iteration count grows linearly with $n/r$, while time per iteration falls up to 7×. Against SGD, the method reaches $\varepsilon = 10^{-2}$ roughly 2.5× faster for $s = n/4$, demonstrating that sketching *plus* the second-order update can lead to significant runtime improvements.

## 5.4 Evaluation on non-synthetic data

We evaluate the unconstrained EAGLE variant on seven low-rank matrices from the SuiteSparse Matrix Collection Davis and Hu (2011) spanning 1100 million coefficients. For each matrix of rank $r$, rows and columns are permuted such that the top-right $r \times r$ block $A$ is full-rank. For compatability with existing CUDA solvers, we reduce the block matrices $B, D$ to a single row, yielding a setting akin to the linear system $wA = B$.

We benchmark EAGLE against three GPU-accelerated baselines implemented in `cupyx.scipy.sparse.linalg`: an SVD solver, conjugate gradients on the normal equations (CG; Hestenes and Stiefel, 1952), and LSMR (Fong and Saunders, 2011). We run each iterative solver until a residual of $10^{-3}\|B\|_F^2$ is reached. Table 2 lists wall-clock times in seconds. First, we observe that EAGLE consistently outperforms the SVD-based solver despite its nominal $\log \kappa$ overhead; we attribute this to its heavy use of efficient, massively parallel matrixmatrix multiplication. Second, EAGLE is competitive with sparse iterative methods—beating LSMR on every matrix and CG on all but one—highlighting EAGLEs practical strength for large, ill-conditioned linear systems. §E.4 corroborates both results for varying residual tolerances.

| Task | EAGLE | SVD | CG | LSMR |
|------|-------|-----|-----|------|
| Maragal_4/Maragal_4 | 0.625 | 0.849 | 1.55 | 2.34 |
| Maragal_5/Maragal_5 | 0.632 | 1.40 | 1.75 | 1.40 |
| Meszaros/pf2177 | 3.43 | 4.23 | 1.75 | 3.51 |
| HB/dwt_1007 | 0.640 | 0.902 | 4.65 | 9.73 |
| HB/bcsstm13 | 0.697 | 0.955 | 3.17 | 1.16 |
| Priebel/162bit | 0.847 | 2.35 | 16.7 | 57.1 |
| Schulthess/N_pid | 0.675 | 1.15 | 1.27 | 1.96 |
| average rank | 1.14 | 2.29 | 2.86 | 3.71 |

Table 2: Runtime in seconds of different numerical solvers on real-world matrix data. EAGLE is a strong competitor for solving linear systems.

## 6 Conclusions

We view transformers as fixed data-to-data transforms whose forward pass exposes emergent algorithms. We fix one low-rank matrix-completion task and explore various regimes. Our proposed rule, EAGLE is a transformer-induced method that emerges as a unifying algorithm in centralized, distributed and sketching regimes, achieving second-order convergence with only $\log \kappa$ (where $\kappa$ is the condition number), $\alpha^{-1}$ (distributed data diversity) or $n/r$ (sketch) slow-downs. Empirically, it outperforms Conjugate Gradient and Gradient Descent by 12 orders-of-magnitude in both iteration complexity and net communication, and matches the behaviour of QR-based solvers up to $\log$ terms.

*Limitations.* Numeric stability beyond $\kappa > 10^8$ is untested. Second, we show one pre-training recipe that yields EAGLE; extensions to non-linear regimes that do so are open.

**Acknowledgements**

This research was supported by the Army Research Office Grant W911NF2110246, AFRL Grant FA8650-22-C1039, and the National Science Foundation grants CPS-2317079, CCF-2007350, DMS-2022446, DMS-2022448, and CCF-1955981.

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

# A   Related Work

**Neural networks as implicit algorithm learners.** Large models can *emerge* classical iterative procedures when trained on synthetic tasks. For transformers this spans gradient descent for least squares (Von Oswald et al., 2023; Ahn et al., 2023), dual gradient descent for optimal transport (Daneshmand, 2026), and temporaldifference planning in RL (Wang et al., 2025). RNNs and MLPs exhibit similar behavior (Siegelmann and Sontag, 1992; Tong and Pehlevan, 2024), though attention yields depth-aligned, inspection-friendly updates (Garg et al., 2022). **Gap.** All of these works uncover *first-order* rules tied to a single regime. **Our advance.** EAGLE is the first *second-order* update that generalizes *unchanged* across centralized, distributed, and sketched settings.

**Data transformers *vs.* parameter updates.** Unlike classical solvers that iterate in *parameter space*, a transformers computation is purely *data-to-data*: weights stay fixed at inference time, and all state lives inside the forward activations. Prior analyses of in-context learning focus on mapping this data flow to gradient descent on hidden parameters (Akyürek et al., 2023; Von Oswald et al., 2023). EAGLE shows a stronger resultan *explicit, second-order data-space projector* that converges quadratically without ever touching model parameters. This clarifies that algorithmic power can emerge *without* an internal parameter update loop.

**In-context learning as regressionrates and limits.** Theoretical analyses cast in-context learning as linear regression on prompt examples (Akyürek et al., 2023; Bai et al., 2023; Min et al., 2021). Transformers so far reproduce gradient descent or SGD, yielding linear rates and the usual $\sqrt{\kappa}$ condition-number dependence. **Our advance.** We extract an *emergent* NewtonSchulz-style update with quadratic convergence and $\log \kappa$ dependence that remains numerically stable under three disparate resource constraints.

**Classical second-order solvers, Nyström, and low-rank approximation.** NewtonSchulz iterations deliver quadratic convergence for matrix inversion (Higham, 1997) but require explicit matrix products and have not been analyzed in noisy, data-dependent regimes. Nyström methods scale kernel learning and matrix completion (Williams and Seeger, 2001; Halko et al., 2011; Gittens and Mahoney, 2016); state-of-the-art variants use QR or Krylov subspace solvers (Saad, 2003; Musco and Musco, 2017). **Gap.** These algorithms are hand-designed for fixed settings and degrade to first-order ($\sqrt{\kappa}$) rates when run under communication or memory limits. **Our advance.** The transformer learns the same NewtonSchulz-type projector that attains Nyström accuracy in only $\log \kappa$ steps and *adapts automatically* to distributed or sketched execution.

**Distributed and sketched solvers.** Block-CG, communication-optimal Krylov methods (Demmel et al., 2013; Hoemmen, 2010) and randomized sketch-and-solve techniques (Clarkson and Woodruff, 2017; Woodruff, 2014) are the de-facto choices at scale, yet remain first-order and $\sqrt{\kappa}$-limited. We introduce a *diversity index* $\alpha$ that predicts our distributed rate and achieves provably fewer rounds when worker subspaces overlap ($\alpha^{-1} \ll \sqrt{\kappa}$).

**Meta-learning across tasks.** Transformers pretrained on heterogeneous corpora behave as meta learners (Min et al., 2021; Bai et al., 2023; Naim and Asher, 2025); prior work, however, stops at behavioral observation. EAGLE offers a mechanistic account: pre-training can imprint a *single* algorithm that flexes with compute, memory, and communication budgets.

**Scope.** This survey is necessarily selective; we focus on works most relevant to the *data-space, second-order* perspective. A broader taxonomy of numerical solvers and in-context phenomena is beyond our scope but complementary to the questions addressed here.

**Positioning.** To our knowledge this is the first demonstration that pre-training alone can induce *one* second-order, matrix-completion rule that (i) bridges centralized, distributed, and sketch regimes, (ii) admits tight non-asymptotic guarantees, and (iii) out-iterates classical Krylov baselines *and* cuts communication under typical ML-scale conditioning.

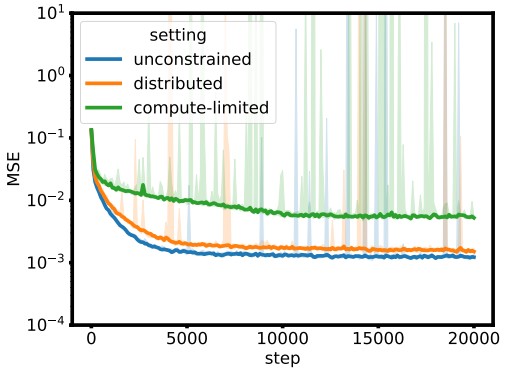

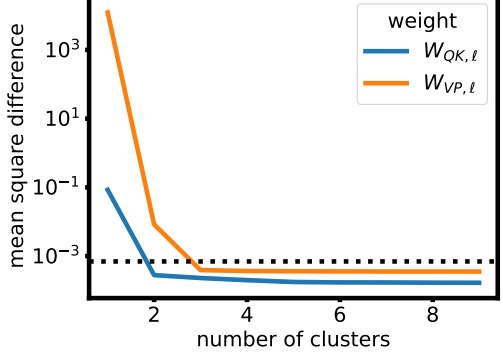

Figure 9: Training loss, median across 10 independent runs.

Figure 10: Prediction difference when sparsifying weights in centralized setting. For reference, the black dotted line marks the trained transformer's mean squared error. Mean across 10 training runs reported.

## B    Methodological Details

### B.1    Nyström Approximation

The Nyström approximation of a block matrix $X = \begin{bmatrix} A & C \\ B & D \end{bmatrix}$ with missing block $D$ is the matrix

$$\hat{X}_* = \begin{bmatrix} A & C \\ B & \hat{D}_* \end{bmatrix},$$

where

$$\hat{D}_* := BA(AA^\top)^\dagger C.$$

Here, $Z^\dagger$ is the Moore-Penrose pseudoinverse of $Z$. Note that this approximation ensures that the rank of the approximate $\hat{X}_*$ is the same as the rank of $A$—in this way, the Nystrom approximation reduces both the rank of $X$, as well as the Frobenius norm of the error. The latter property can be seen from the fact that the approximation just amounts to a joint execution of ordinary least squares. Indeed, if we treat the $i$th row of $B$, $B^i$ as a 'response variable' and the $j$th column of $C, c^i$ as a query point, then

$$(\hat{D}_*)_{ij} = B^i A(AA^\top)^\dagger c^j$$

is precisely the estimate when one regreeses the data $(A, B^i)$ onto $c^j$. For this reason, we may also interpret the method as computing the estimation parameters

$$W_* := BA(AA^\top)^\dagger \in \mathbb{R}^{d' \times d},$$

and then computing the estimate

$$\hat{D}_* = W_* C,$$

much as in ordinary linear regression. This value $W_*$ will appear later in our theoretical analyses.

We note that the approximation above is defined whether the matrix $X$ is noise-corrupted or not. In the presence of noise, it inherits many of the statistical properties of linear regression when we assume that the noise is restricted to the 'response variables' $B, D$. As a special case, if the rank of $X$ is equal to that of $\begin{bmatrix} A & C \end{bmatrix}$, then in fact it holds that $\begin{bmatrix} B & D \end{bmatrix} = W_* \begin{bmatrix} A & C \end{bmatrix}$, i.e., the Nyström approximation is exact. We will sometimes refer to this as the 'noiseless' case.

### B.2    Training specifications

The transformer is trained using the Adam optimizer with a constant learning rate of 0.001 and a batch size of 1024 for 20,000 iterations. To stabilize training, gradient clipping with a 2-norm threshold of 0.1 is applied at each step. At every iteration, a new batch is independently sampled

as detailed in Section 3, ensuring that the model never sees the same data twice during training or inference. The block masking pattern remains fixed throughout. We observe occasional spikes in the training loss (Figure 9) and interpret those as occasional failures of the learned solver to converge, typically occurring when $\|A\|_2$ attains untypically large values. We recall that, under our data sampling procedure, $\|A\|_2$ is random and unbounded, allowing for such outlier cases.

## B.3   Runtime of attention map

Recall that the dominant computational cost in a transformer layer arises from the attention mechanism:

$$\text{Attn}(Z) = \big(ZW_Q(ZW_K)^\top \odot M\big)ZW_VW_P^\top,$$

where $Z \in \mathbb{R}^{(d+d')\times(n+n')}$ is the input matrix and $W_Q, W_K, W_V, W_P \in \mathbb{R}^{n\times k}$ are learned projection matrices. Assuming that the size of the submatrix $D$ remains constant, we have $d' = \Theta(1)$ and $n' = \Theta(1)$.

The computation of $ZW_Q$, $ZW_K$, and $ZW_V$ requires $\Theta(ndk)$ time. The inner product and masking operation $(ZW_Q)(ZW_K)^\top \odot M$ incurs a cost of $\Theta(d^2k)$, and the subsequent multiplication with $ZW_VW_P^\top$ requires an additional $\Theta(d^2k + ndk)$. Thus, the total time complexity of the attention computation is $\Theta(d^2k + ndk)$.

For the unconstrained case $k = n$, this yields a cost of $\Theta(nd^2 + dn^2)$. Therefore, selecting $k < n$ can significantly reduce computational overhead.

Of course, in our recovered algorithms, we can 'precompute' $W_QW_K^\top$ and $W_VW_P^\top$, and these are further scaled versions of block-identity matrices, and so the computation of $ZW_Q$ etc. can be avoided. This reduces the cost to $O(d^2k)$, which for the unconstrained setting of $k = n$ works out to $O(d^2n)$.

In the sketched versions of the iterations, these matrices become nontrivial again, and their multiplication must be incorporated into the accounting of the costs of the recovered algorithm. Of course, in this regime, $k = r \ll n$, so the domainting cost is $O(ndr)$.

## B.4   Details on setup for distributed setting

We design the transformer to emulate a distributed algorithm by constraining the attention mechanism. Specifically, the query, key, and value matrices are restricted to data local to a single machine, and due to symmetry across machines, a single set of transformation is shared across all attention heads. At every layer, the input data is partitioned as:

$$Z_k = \begin{bmatrix} \cdots & X_k^\mu & X_k^{\mu+1} & \cdots \end{bmatrix} \in \mathbb{R}^{(d+d')\times M(n+n')}, \text{ where } X_k^\mu = \begin{bmatrix} A_k^\mu & C_k \\ B_k^\mu & D_k \end{bmatrix}, \mu \in [1:M]$$

with blocks $A^\mu \in \mathbb{R}^{n\times d}$ and $B^\mu \in \mathbb{R}^{n'\times d}$ distributed across machines, while the shared blocks $C \in \mathbb{R}^{n\times d'}$ and $D \in \mathbb{R}^{n'\times d'}$ are replicated identically in each $X^\mu$. This setup reflects common scenarios where the missing block $D$ is significantly smaller than the full matrix $A$, often remaining constant in size (e.g., $d' = n' = 1$ in regression), making the distributed partitioning both efficient and natural.

The projection matrix after attention is unconstrained, enabling unrestricted communication between heads and implicitly modeling a fully connected communication topology. This architectural design encourages local, machine-specific computation in the attention mechanism, while allowing for global coordination in the projection step. Empirically, we find that the model leverages this flexibility efficiently: although capable of learning full data sharing, it consistently limits itself to communication scaling with $O(|D|)$ bits.

We find that transposing the per-machine data $X^\mu$ is necessary in the distributed setup. That is, we give the transformer inputs $Z_k = \begin{bmatrix} \cdots & X_k^{\mu,\top} & X_k^{\mu+1,\top} & \cdots \end{bmatrix}$ and train it on the mean-squared error to $D^\top$.

| layer | U$\times 10^4$ | D$\times 10^4$ | C-L$\times 10^4$ |
|---|---|---|---|
| 0 | 0.00 | 0.00 | 0.00 |
| 1 | 2.36 | 0.62 | 0.01 |
| 2 | 5.27 | 1.40 | 0.02 |
| 3 | 3.70 | 1.28 | 0.13 |
| 4 | 2.16 | 1.32 | 0.14 |

Table 3: Differences across iterations between transformer and extracted algorithm (squared Frobenius norm divided by size of $Z$) in the unconstrained (U), distributed (D), and compute-limited (C-L) settings. Mean *times* $10^4$ across 10 seeds is reported.

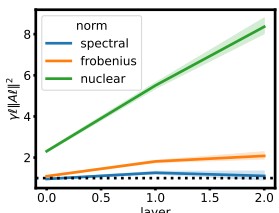

Figure 11: The transformer learns to normalize the batch-maximum spectral norm of $A_\ell$ observed during training. The plot reports $\alpha_\ell^1 \alpha_\ell^2 \max_{b \in [B]} \|A_\ell^{(b)}\|^2$, where the maximum is taken over each batch.

### B.5 Details on algorithm extraction

We derive closed-form iterative updates from the trained transformer weights $W_{QK,\ell}$ and $W_{VP,\ell}$. This involves thresholding and clustering the weight matrices in both the unconstrained and distributed settings:

- The sparsification threshold $\tau$ is set dynamically as $\tau = 1.5 \cdot \|W\|_1 / \#\text{elements}$, accounting for varying scales across weight matrices. The constant 1.5 is selected empirically and is not further tuned.

- The number of quantization levels is determined by analyzing the mean prediction error induced by weight sparsification (see Figure 10). To maintain performance comparable to the original transformer, we cluster $W_{QK,\ell}$ into 2 values and $W_{VP,\ell}$ into 3.

In the compute-limited setting, a sketch matrix is extracted and its structural properties analyzed (see Section 4). Parameter choices for the update steps are abstracted through empirically observed scaling laws. Specifically, we identify $\alpha_1 \alpha_2 \approx 1/\|A\|_2^2$, as supported by results in the main text. Furthermore, we approximate $\alpha_1 \alpha_3 \approx 1.92 \cdot \alpha_1 \alpha_3$, where the resulting squared prediction error is $3.5 \times 10^{-4}$ (averaged over 10 training runs), remaining below the baseline error of approximately $1 \times 10^{-3}$.

## C Emergent Algorithm

### C.1 Numerical evidence for faithful extraction

Table 3 shows the difference between the transformer embeddings and abstracted iterations for every transformer layer. The largest error of $6 \times 10^{-4}$ is still substantially smaller than the transformer's overall predictive performance after 4 layers (around $1 \times 10^-3$. This shows that the abstracted iteration truthfully reflects the numerical procedure implemented by the transformer.

Figure 11 shows the product $\alpha_\ell^1 \alpha_\ell^2 \max_{b \in [B]} \|A_\ell^{(b)}\|^2$ for difference choices of norms. In this expression, the maximum is taken over a training bach of size $B$. The plot reveals that the transformer scales the constant $\alpha_\ell^1 \alpha_\ell^2$ with the maximum spectral norm of $A_\ell$ expected at layer $\ell$ during training.

### C.2 Transformer behaviour under data noise

In this section, we demonstrate that the emergent behavior identified in Section 4 remains robust under moderate data noise with variance $\sigma^2 = 0.01$.

Figure 13 compares the learned weights of a representative transformer trained on noiseless data and on data corrupted with noise. Additionally, Figure 14 reports key diagnostic quantities across layers. In both analyses, the observed behavior closely aligns with that of the noiseless case, indicating stability of the learned algorithm under modest perturbations.

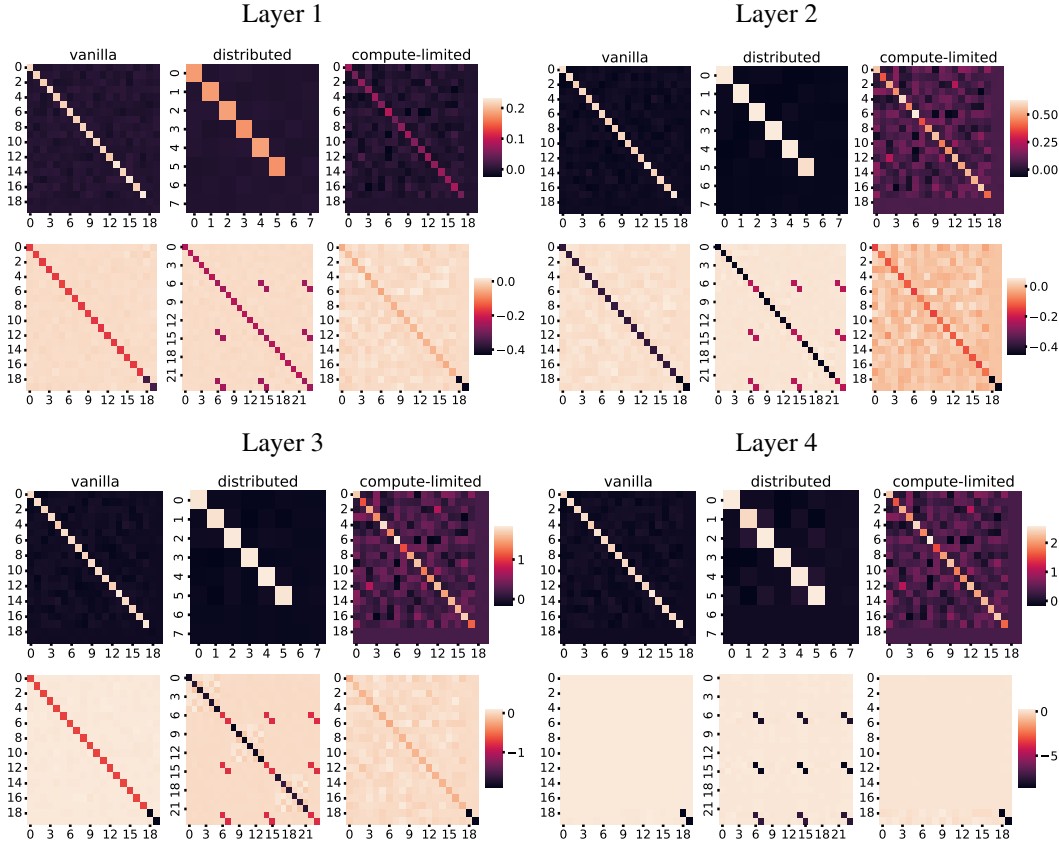

Figure 12: Transformer weights by layer, averaged over 10 training runs. The top panels display the weight products $W_{QK,\ell}$, and the bottom panels show $W_{VP,\ell}$. Prior to averaging, sign ambiguities were resolved by aligning the dominant coefficients: large entries in $W_{QK,\ell}$ were set to be positive, and those in $W_{VP,\ell}$ to be negative.

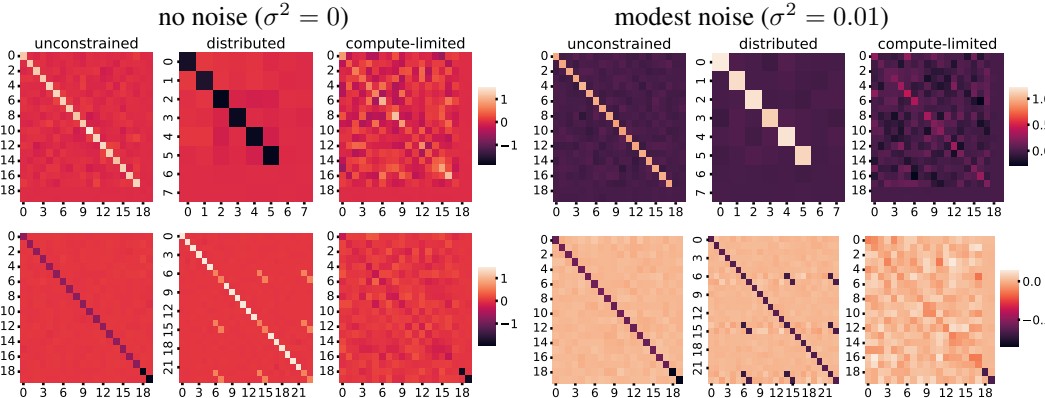

Figure 13: Comparison of learned model weights without noise vs. with modest data noise on a representative training run. The top row shows $W_{QK,\ell}$ and the bottom row shows $W_{VP,\ell}$.

## C.3 Transformer behaviour across training runs

We show that the emergent behavior identified in Section 4 is robust across training runs with varying sources of randomness, including initialization and training data. Figure 12 displays transformer weights averaged over 10 independent runs, reproducing the characteristic patterns reported in Section 4. Additionally, the sketching sub-matrix in the compute-limited setting is, on average, close to the identity, further supporting out interpretation as a random orthonormal sketch.

## C.4 From weights to updates

We derive the blockwise update rule implied by the abstracted weight parametrization

$$W_{QK,\ell} = \begin{bmatrix} \alpha_1 I & 0 \\ 0 & 0 \end{bmatrix} \quad \text{and} \quad W_{VP,\ell} = \begin{bmatrix} \alpha_2 I & 0 \\ 0 & \alpha_3 I \end{bmatrix}$$

and show that it recovers the update structure presented in Section 4 for the unconstrained setting. Recall that the layer representation takes the block form

$$Z_\ell = \begin{bmatrix} A_\ell & C_\ell \\ B_\ell & D_\ell \end{bmatrix}$$

and a single-head transformer layer performs the update

$$Z_{\ell+1} = Z_\ell + \left( Z_\ell W_{QK,\ell} Z_\ell^\top \odot M_\ell \right) Z_\ell W_{VP,\ell}. \tag{4}$$

Substituting the block structure into the attention term yields

$$Z_\ell W_{QK,\ell} Z_\ell^\top = \begin{bmatrix} \alpha_1 A_\ell A_\ell^\top & \cdot \\ \alpha_1 B_\ell A_\ell^\top & \cdot \end{bmatrix}$$

where the entries marked $\cdot$ are not needed due to masking. Applying the attention mask,

$$Z_\ell W_{QK,\ell} Z_\ell^\top \odot M_\ell = \begin{bmatrix} \alpha_1 A_\ell A_\ell^\top & 0 \\ \alpha_1 B_\ell A_\ell^\top & 0 \end{bmatrix}$$

Further, we compute

$$Z W_{VP,\ell} = \begin{bmatrix} \alpha_2 A_\ell & \alpha_3 C_\ell \\ \alpha_2 B_\ell & \alpha_3 D_\ell \end{bmatrix}.$$

Multiplying these terms and adding to $Z_\ell$, we obtain the update:

$$Z_{\ell+1} = Z_\ell + \begin{bmatrix} \alpha_1 \alpha_2 A_\ell A_\ell^\top A_\ell & \alpha_1 \alpha_3 A_\ell A_\ell^\top C_\ell \\ \alpha_1 \alpha_2 B_\ell A_\ell^\top A_\ell & \alpha_1 \alpha_3 B_\ell A_\ell^\top C_\ell \end{bmatrix}.$$

This clean separation of updates across the blocks $A_\ell$, $B_\ell$, $C_\ell$, and $D_\ell$ a posteriori motivates the choice of block decomposition of $Z_\ell$ used in our notation throughout the model layers.

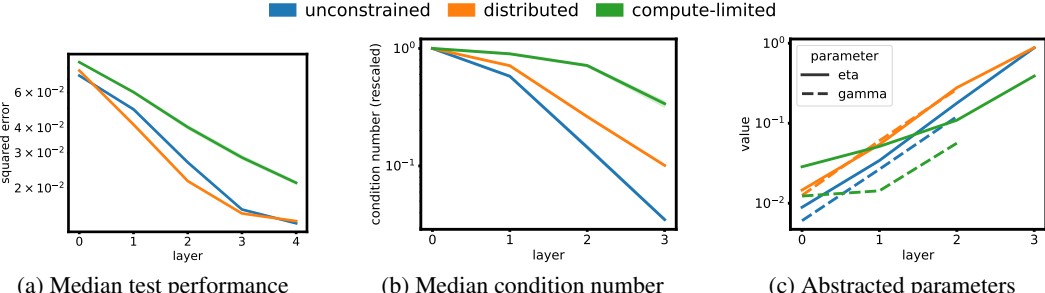

(a) Median test performance      (b) Median condition number      (c) Abstracted parameters

Figure 14: The trained transformer solves matrix completion (modest noise, $\sigma^2 = 0.01$) with a unified algorithm over all three computational settings. The evolution of key quantities throughout the transformer layers illustrate the remarkable similarity between the latent algorithms. Mean across 10 training seeds is reported.

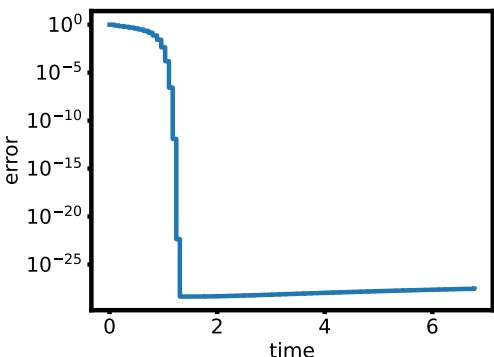
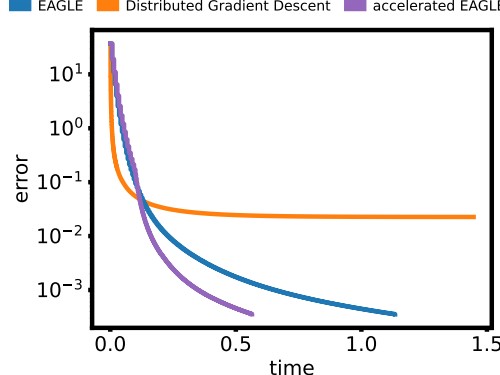

Figure 15: Convergence of EAGLE adapted to the estimation problem: unconstrained setting, with parameters $n = d = 240$, $n' = d' = 2$, $\kappa(A) = 100$. Mean over 50 runs is reported.

Figure 16: The accelerated EAGLE for distributed computing largely outperforms EAGLE in runtime. In this data generation, the data diversity $\alpha \approx 0.1$ is relatively low. Mean over 50 runs is reported, $n = d = 240$, $n' = d' = 2$, $\kappa(A) = 100$

### C.5 Interpreting the emergent algorithm

Prior work interprets EAGLE as gradient descent with preconditioning (Von Oswald et al., 2023; Ahn et al., 2023; Vladymyrov et al., 2024). In particular, these works frame the method as comprising two distinct phases: a preconditioning step involving only the updates to $A$ and $B$ (this is achieved by setting $\gamma = 0$), followed by a single gradient descent step using the data $(A_\ell, B_\ell, C_0)$. This perspective treats the conditioning phase as auxiliary and external to the actual optimization step.

However, this interpretation does not accurately reflect the behavior of the trained transformer. The model does not explicitly separate conditioning and update phases; instead, it interleaves them continuously throughout the computation. Rather than implementing classical preconditioning followed by gradient descent, the learned procedure functions as a continuous conditioning mechanism that shapes the dynamics of the optimization process at every layer.

Moreover, this prior viewpoint underemphasizes the internal structure and complexity of the conditioning dynamics themselves. In fact, the iteration bears a close resemblance to the Newton–Schultz method for matrix inversion, suggesting a fundamentally different mechanism at play.

To make this connection explicit, consider the unconstrained version of the iteration. Assume $\|A\|_2 = 1$, and set $\eta = 1$, $\gamma = 3$, as in the statement of Theorem 1. Then, re-normalize $\bar{A}_\ell = A_\ell / \|A_\ell\|_2$ at every iteration. The update to the iterates $\bar{A}_\ell$ in this setup can then be expressed in the compact form

$$\bar{A}_{\ell+1} = \frac{1}{2}(3I - \bar{A}_\ell \bar{A}_\ell^\top)\bar{A}_\ell,$$

which is exactly the Newton–Schultz iteration for computing the matrix sign function, as presented in (Higham, 2008, Section 5.3). This reformulation reveals that the driving force behind the iteration is more accurately characterized as implicit matrix inversion rather than traditional gradient descent updates.

## D   Parameter Estimation

Beyond the standard prediction task in which the hidden block $D$ is to be recovered, one may also be interested in estimating the matrix $W^\star = BA(AA^\top)^\dagger$. In the least-squares variant of our Nyström formulation—where $D$ reduces to a scalar $1 \times 1$ block—this corresponds to solving the classical least-squares problem. While it is straightforward to recover $W^\star$ by setting $C = I$ in EAGLE (Algorithm 1), so that the prediction becomes $BA(AA^\top)^\dagger C = BA(AA^\top)^\dagger = W^\star$, this approach is inefficient in terms of memory and runtime. It is thus natural to ask whether the estimation task can be achieved more efficiently, with reduced overhead.

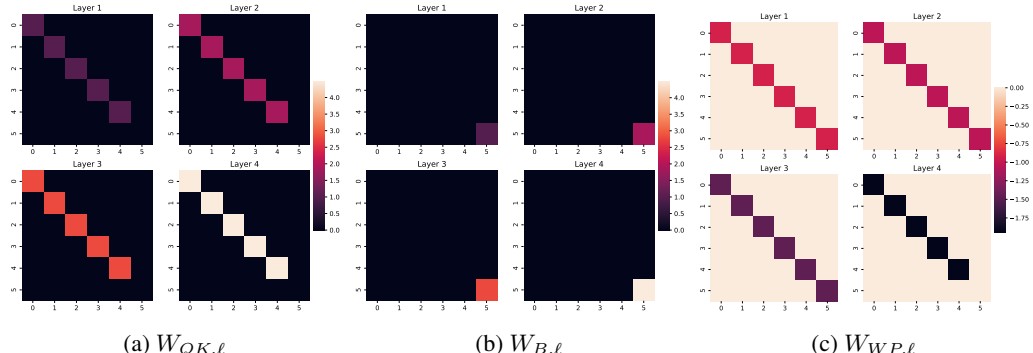

(a) $W_{QK,\ell}$          (b) $W_{B,\ell}$          (c) $W_{WP,\ell}$

Figure 17: Learned transformer weights in the estimation setting. The transformer architecture is modified and a bias term is introduced: $Z_{\ell+1} = Z_\ell + \big((Z_\ell W_{QK,\ell} + W_{B,\ell})Z_\ell^\top \odot M_\ell\big)Z_\ell W_{VP,\ell}$ where $W_B \in \mathbb{R}^{n \times d}$ is a rank-2 bias term whose first $n-1$ rows are identical by design.

To investigate this question, we consider an adapted transformer architecture trained on a matrix completion problem posed as

$$Z = \begin{bmatrix} A^\top & B^\top \\ ? & 0 \end{bmatrix},$$

where the transformer is tasked with filling the missing block "?" with $W^\star = BA(AA^\top)^\dagger$. We find that a minor architectural modification is necessary to achieve competitive performance on this estimation task. Specifically, we introduce two biases to the query transformation. The first bias is applied when updating the first $n$ tokens, and the second is applied when updating the last $n'$ tokens.

We train a transformer on the least-squares setting (i.e., with $D \in \mathbb{R}^{1 \times 1}$) and extract the learned weights, shown in Figure 17. This leads to a modified estimation version of EAGLE, defined by the following iterative updates:

$$
\begin{aligned}
A_{\ell+1} &= A - \eta\rho_\ell A_\ell A_\ell^\top A_\ell, \\
B_{\ell+1} &= B - \eta\rho_\ell B_\ell A_\ell^\top A_\ell, \\
W_{\ell+1} &= W_\ell - \gamma\rho_\ell (W_\ell A_\ell - B_\ell)A_\ell^\top.
\end{aligned}
$$

These updates can be adapted to all computational settings considered in the main body. Importantly, this iteration is theoretically equivalent to the original EAGLE algorithm in the sense that $W_k C = D_k$, where $W_k$ evolves according to the update rule above and $D_k$ evolves according to EAGLE. This equivalence can be formally shown by expanding the recursive updates and applying a telescoping argument, as in the proof in § F.

Figure 15 provides empirical evidence that the proposed estimation iteration (with the same parameter setup $\rho_\ell = 1/(3\|A_\ell\|_2^2)$, $\eta = 1$, $\gamma = 3$) converges as expected and recovers the same second order convergence as EAGLE.

## E    Further Details on Evaluation of EAGLE

### E.1    Details on the implementation of baselines

We provide additional implementation details for the baseline methods used in comparison with EAGLE. All three implementations compute an approximation of $X^\star = BA^\dagger$ and return $D \approx X^\star C$.

**Exact closed-form baseline (`np.linalg.lstsq`)**    This native numpy impementation of the a least-squares solver is base on a highly optimized LAPACK routine. It determines solution to the over/under-determined linear system $\min_X \|XA - B\|_F^2$. The closed-form minimiser is $X^\star = BA(AA^\top)^\dagger$. It returns $D = X^\star C$.

**Gradient Descent.**   We minimize the reconstruction loss

$$f(X) = \tfrac{1}{2} \sum_{i=1}^{m} \|XA - B\|_F^2.$$

*Parameters.* The learning rate is set to $\eta_\ell = 1/(\sigma_{\max}^2(A))$, ensuring monotonic decrease of $f$. Spectral norms are estimated via two power iterations, which incur negligible memory and runtime overhead. To avoid instabilities caused by near-singular matrices, we add a fixed ridge penalty $\lambda = 10^{-3}$ when $A$ is low-rank, modifying the gradient as $G \leftarrow G + \lambda X$. When $A$ is full rank, we use $\lambda = 0$.

*Iteration and Variants.* The method extends naturally to distributed or stochastic data acess variants. Each worker computes either exact or column-sketched gradients ($\tilde{A}_\ell^i$ is the data on worker $i$ with subsampled columns) which are aggregated on a central node, followed by a synchronous gradient step ($\eta_k = 1/\max_i(\sigma_{\max}^2(\tilde{A}_k^i))$). This yields the following iterative updates:

$$G_k = \sum_{i=1}^{m}(X_k\tilde{A}_k^i - \tilde{B}_k^i)\tilde{A}_k^{i,\top} + \lambda X_k \qquad \text{(gradient + ridge)}$$

$$X_{k+1} = X_\ell - \frac{\eta_k}{m}G_k \qquad\qquad\qquad \text{(synchronous update)}$$

Training stops when the iterate $X_\ell$ reaches a predefined tolerance or after a fixed maximum number of iterations.

**Conjugate Gradient Method.**   We implement the Conjugate Gradient (CG) method to solve the normal equations arising in Nyström estimation. The objective is to find $X$ minimizing the empirical loss $\frac{1}{2}\sum_{i=1}^{m}\|XA - B\|_F^2$. We define the Gram matrix $G = AA^\top$ and the right-hand side $B = BA^\top$, and solve the system $XG = B$ using CG.

*Iterations.* The standard CG updates are as follows:

$$R_k = B - X_kG, \qquad \text{if } k = 0: \ P_0 = R_0, \ rr_0 = \langle R_0, R_0 \rangle$$

$$\alpha_k = rr_k/\langle P_k, \ P_kG \rangle, \qquad X_{k+1} = X_k + \alpha_k P_k$$

$$R_{k+1} = R_k - \alpha_k P_kG, \qquad rr_{k+1} = \langle R_{k+1}, R_{k+1} \rangle$$

$$\beta_k = rr_{k+1}/rr_k, \qquad P_{k+1} = R_{k+1} + \beta_k P_k$$

The algorithm halts when $\|X_kC - X_{k-1}C\|_F$ falls below a pre-specified tolerance or after a fixed number of iterations.

*Limitations.* The CG method is not directly suited for distributed or stochastic settings. In distributed environments, computing $G$ and $B$ requires sharing all local data across machines, incurring communication costs that exceed the desired $O(d)$ complexity. In stochastic settings, CG requires a consistent system matrix at every iteration to guarantee convergence, which is not available when data is sampled independently at each step. Empirically, CG under stochastic updates exhibits unstable and non-convergent behavior.

### E.2   Accelerated EAGLE for distributed setting

In the distributed setting, convergence is impeded by low data diversity, characterized by $\alpha \ll 1$. Recall that the continuous conditioning performed by EAGLE operates exclusively on per-machine data. As a result, the condition number of the global matrix $A$ is lower bounded by $\alpha^{-1}$, even if the data on each individual machine is perfectly conditioned.

Once this bottleneck is reached, further updates to $A^\mu$the most computationally intensive component of EAGLEcease to be effective. This motivates the question of whether runtime can be reduced by terminating updates to $A^\mu$ and $B^\mu$ once per-machine conditioning has converged.

To this end, we implement an accelerated version of EAGLE for the distributed setting. At each iteration, we evaluate the criterion $\min_\mu \|I - A_\ell^{\mu,\top}A_\ell^\mu\|_F^2 < 10^{-10}$, and halt updates to $A^\mu$ and $B^\mu$

for all $\mu$ once this threshold is satisfied. This condition ensures that per-machine data is sufficiently well-conditioned.

Figure 16 reports the runtime performance of this accelerated variant. As expected, the modified version achieves faster execution compared to standard EAGLE. We note that the number of iterations required remains unchanged; the acceleration only affects wall-clock efficiency. The theoretical guaranteed of the accelerated EAGLE remain unchanged as we ensure that $\kappa(A_\ell^\mu) = 1$ (to machine precision) before freezing the updates to $A^\mu$ and $B^\mu$.

### E.3 Ablations for EAGLE

**Data distribution.** We perform ablations on various real-world data settings. As in the main body, we place ourselvels in the setting $n = d = 240$, $n' = d' = 2$ and $r = 240$.

*SVD construction.* Sample $A_{ij}, B_{ij}$ orthogonal and set $Z = A\Sigma B^\top/\sqrt{r}$ with $\Sigma$ diagonal with entries log-uniform in $[1e - 2, 1]$. This is the reference distribution in the simulation part.

*Gaussian.* Sample $A_{ij}, B_{ij} \sim \mathcal{N}(0, 1)$ and set $Z = AB^\top/\sqrt{r}$. Gaussian data is a widely made modeling assumption, and this is the distribution the transformer was trained on.

*Student–$t_\nu$ factors.* Sample $A_{ij}, B_{ij} \sim t_\nu/\sqrt{\nu/(\nu - 2)}$ for $\nu = 4$ and set $Z = AB^\top/\sqrt{r}$. Heavy-tailed entries create sporadic outliers, checking that the solver is stable beyond sub-Gaussian data.

*Correlated Gaussian factors.* Draw each column of $A$ and $B$ from $\mathcal{N}(0, \Sigma)$ with $\Sigma_{ij} = \rho^{|i-j|}$ ($\rho = 0.8$). Strong row/column correlations are typical in spatio-temporal panels and challenge methods assuming independent observations.

*Sparse Rademacher factors.* With sparsity $p = 0.1$, take $A_{ij} = s_{ij}\xi_{ij}/\sqrt{p}$ where $\xi_{ij} \sim \text{Bernoulli}(p)$, $s_{ij} \in \{\pm 1\}$ (analogous for $B$), then $Z = AB^\top/\sqrt{r}$. Entry-wise sparsity yields highly *coherent* singular vectorsa worst-case regime for many theoretical guarantees.

*Block / clustered factors.* Assign each of the $d$ rows to one of $k$ clusters ($k = 5$), $A \in \{0, 1\}^{d \times k}$ is one-hot, and $B \in \mathbb{R}^{n \times k}$ holds cluster centroids; set $Z = AB^\top/\sqrt{k}$. Produces piece-wise-constant structure seen in recommender systems, violating global incoherence yet preserving low rank.

**Larger systems.** We increase the size of the data matrix $X$ to $n = d = 1920$ and $n' = d' = 2$; results are shown in Figure 20. These experiments further reveal that the Conjugate Gradient method does not converge reliably when applied to low-rank matrices. As a result, its performance is omitted from the main results in the low-rank setting.

**Data noise.** To assess robustness to noise, we add moderate i.i.d. Gaussian noise to the matrix $X$ and evaluate EAGLE in all three computational settings; results are presented in Figure 21. We observe that EAGLE exhibits the same form of implicit regularization during early iterations that is well-documented for gradient descent. Across all settings, EAGLE with early stopping achieves performance comparable to gradient descent, recovering regularized solutions that are more stable than the Nyström solution Additionally, we note that the addition of Gaussian noise to $X$ inadvertently improves the conditioning of the matrix $A$, effectively reducing $\kappa(A)$.

**Sensitivity to step sizes.** To evaluate the sensitivity of EAGLE to the choice of step sizes, we conduct a series of controlled experiments in the unconstrained setting, using a small-scale configuration with $n = d = 15$, $n' = d' = 2$, and rank 15.

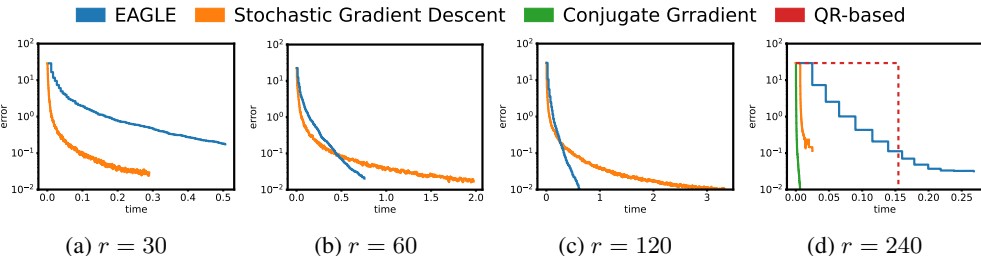

    (a) $r = 30$         (b) $r = 60$         (c) $r = 120$         (d) $r = 240$

Figure 18: In the stochastic data access regimea variant of the compute-limited settingEAGLE outperforms SGD in wall-clock time for sketch sizes $r \geq n/4$.

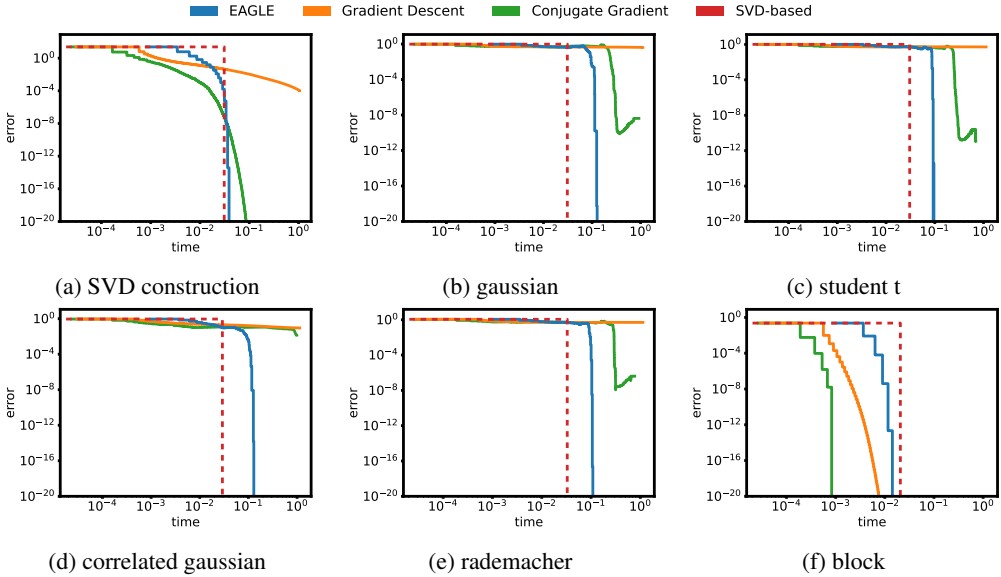

Figure 19: Performance with different data distribution, $n = d = 240$, $n' = d' = 2$, rank 240.

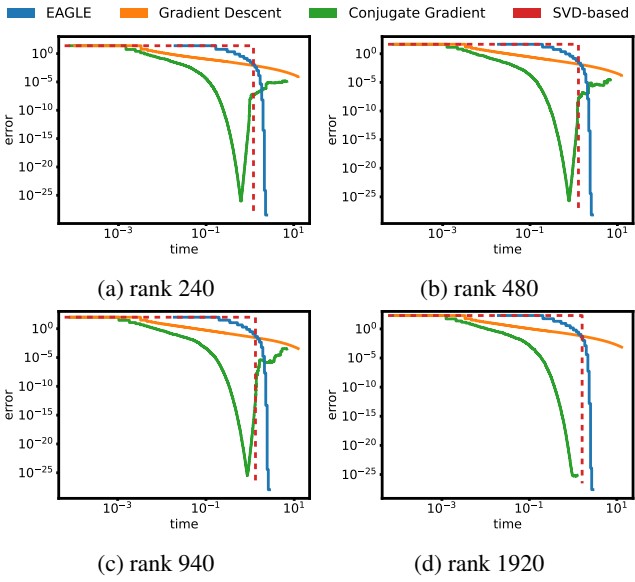

Figure 20: Performance on larger problems, $n = d = 1920$, $n' = d' = 2$.

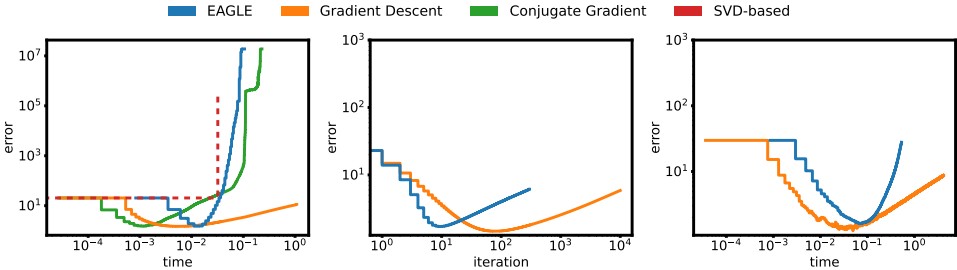

Figure 21: Performance with i.i.d. Gaussian noise (standard deviation $\sigma = 0.01$) in the unconstrained (left), distributed (center) and stochastic data access (right) settings.

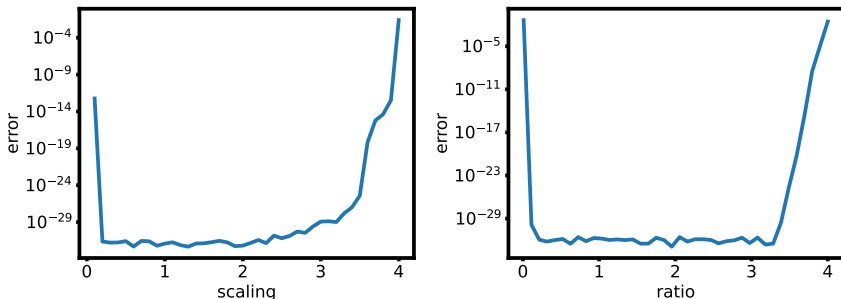

Figure 22: Best performance of EAGLE in 200 iterations as the step sizes are varied. Left: $\gamma$ and $\eta$ are both scaled by a common factor. Right: $\gamma$ is scaled by $\sqrt{\text{ratio}}$ while $\eta$ is scaled by $1/\sqrt{\text{ratio}}$ such that $\gamma/\eta$ is scaled by ratio.

| Task | Size | Nonzeros | rank | conditioning |
|------|------|----------|------|--------------|
| Maragal_4/Maragal_4 | $1964 \times 1034$ | 1.3 % | 801 | $1.38 \times 10^7$ |
| Maragal_5/Maragal_5 | $4654 \times 3320$ | 0.6 % | 2147 | $1.22 \times 10^6$ |
| Meszaros/pf2177 | $9728 \times 10178$ | 0.031% | 9662 | $2.19 \times 10^3$ |
| HB/dwt_1007 | $1007 \times 1007$ | 0.85% | 1000 | $2.97 \times 10^3$ |
| HB/bcsstm13 | $2003 \times 2003$ | 0.53% | 1241 | $2.70 \times 10^4$ |
| Priebel/162bit | $3606 \times 3597$ | 0.29% | 3460 | $1.68 \times 10^4$ |
| Schulthess/N_pid | $3625 \times 3923$ | 0.057% | 2048 | $2.47 \times 10^2$ |

Table 4: Relevant characteristics for each SuiteSparse Matrix Collection task.

First, we scale both step sizes $\gamma$ and $\eta$ by a common factor, as shown in Figure 22 (left). This setup models scenarios in which the estimate of the scaling parameter $\rho_\ell = 1/\|A_\ell\|_2^2$ is inaccurate. The results indicate that scaling $\gamma$ and $\eta$ jointly by a factor in the range $[0.2, 2]$ has minimal effect on performance.

Second, we vary the ratio $\gamma/\eta$ to test the algorithms robustness to discrepancies between these two parameters, as illustrated in Figure 22 (right). We find that EAGLE remains stable over a wide range of ratios; specifically, varying $\gamma/\eta$ within $[0.2, 2]$ has little impact on overall performance.

### E.4   Evaluation on SuiteSparse Matrix Collection data

Table 4 summarizes the characteristics of all evaluation tasks. Moreover, figure 23 gives the runtime of common linear solvers on real-world data as a function of the target residual. We observe three trends. First, EAGLE outperforms the SVD-based solver even when very high accuracies are required. Second, as the theory suggests, the runtime of gradient-based methods out scales that of EAGLE when very high accuracies are required. Finally, we observe a tradeoff where gradient based solvers can outperform EAGLE for large, sparse and well-conditioned systems such as Meszaros/pf2177.

## F   Theoretical Analyses

### F.1   Analysis of the Unconstrained or Centralized Iteration

We begin by analysing the behaviour of Algorithm 1 in the 'centralized' setting of $S = I_n, M = 1$. For simplicity, let us set $\eta_\ell = \eta\|A_\ell\|_2^{-2}$, and $\gamma_\ell = \gamma\|A_\ell\|_2^{-2}$—we will incorporate the specific

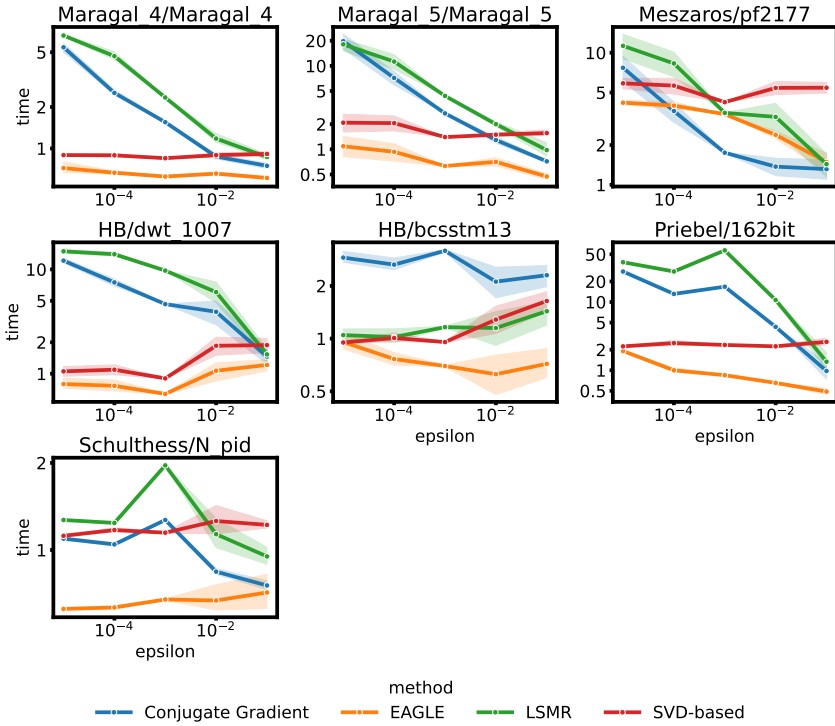

Figure 23: Runtime of linear solvers on real-world data as a function of the target residual $\epsilon \|B\|_F^2$.

structure of $\eta, \gamma$ later. Then we can write the iterations as

$$
\begin{aligned}
A_{\ell+1} &= A_\ell(I - \eta_\ell A_\ell^\top A_\ell), \\
B_{\ell+1} &= B_\ell(I - \eta_\ell A_\ell^\top A_\ell), \\
C_{\ell+1} &= (I - \gamma_\ell A_\ell A_\ell^\top)C_\ell, \\
D_{\ell+1} &= D_\ell + \gamma_\ell B_\ell A_\ell^\top C_\ell.
\end{aligned}
\tag{5}
$$

Of course, we set $A_0, B_0, C_0$ to be the observed blocks of the input matrix $X$, and $D_0 = 0$.

***Useful Definitions.*** We will analyze these iterations through the follow two objects

**Definition 2.** For $\ell \geq 0$, define the 'signal energy' $\mathcal{E}_\ell$, and the 'signal correlation' $\mathcal{V}_\ell$ as

$$
\mathcal{E}_\ell := A_\ell A_\ell^\top \in \mathbb{R}^{d \times d} \text{ and } \mathcal{V}_\ell := B_\ell A_\ell^\top \in \mathbb{R}^{d' \times d}.
$$

We further note the critical relationship that

$$
B_0 = W_* A_0 \implies \mathcal{V}_0 = W_* \mathcal{E}_0,
$$

where $W_*$ is the Nyström parameter for the data matrix $X$ (see §B.1. An immediate consequence of the iterations above is the following dynamics.

**Lemma 3.** *Under the iterations of (5), we have the following dynamics*

$$
\begin{aligned}
\mathcal{E}_{\ell+1} &= \mathcal{E}_\ell(I - \eta_\ell \mathcal{E}_\ell)^2 \\
\mathcal{V}_{\ell+1} &= \mathcal{V}_\ell(I - \eta_\ell \mathcal{E}_\ell)^2 \\
C_{\ell+1} &= (I - \gamma_\ell \mathcal{E}_\ell)C_\ell \\
D_{\ell+1} &= D_\ell + \gamma_\ell \mathcal{V}_\ell C_\ell,
\end{aligned}
\tag{6}
$$

*where $I$ is the $d \times d$ identity matrix.*

*Proof.* The $D_\ell, C_\ell$ iterations follow immediately by definition. For the former iterations, first observe that

$$
\begin{aligned}
\mathcal{V}_{\ell+1} = B_{\ell+1} A_{\ell+1}^\top &= B_\ell (I - \eta_\ell A_\ell^\top A_\ell)(I - \eta_\ell A_\ell^\top A_\ell) A_\ell^\top \\
&= B_\ell (I - \eta_\ell A_\ell A_\ell^\top)(A_\ell^\top - \eta_\ell A_\ell^\top A_\ell A_\ell^\top) \\
&= B_\ell (I - \eta_\ell A_\ell A_\ell^\top) A_\ell^\top (I - \eta_\ell A_\ell A_\ell^\top) \\
&= B_\ell A_\ell^\top (I - \eta_\ell A_\ell A_\ell^\top)(I - \eta_\ell A_\ell A_\ell^\top) \\
&= \mathcal{V}_\ell (I - \eta_\ell \mathcal{E}_\ell)^2,
\end{aligned}
$$

where the first few inequalities arise by multiplying $A_\ell^\top$ from the right, and then factoring it out by the left, and the final is by definition. Similarly,

$$
\begin{aligned}
\mathcal{E}_{\ell+1} = A_{\ell+1} A_{\ell+1}^\top &= A_\ell (I - \eta_\ell A_\ell^\top A_\ell)(I - \eta_\ell A_\ell^\top A_\ell) A_\ell^\top \\
&= A_\ell A_\ell^\top (I - \eta_\ell A_\ell A_\ell^\top)^2 \\
&= \mathcal{E}_\ell (I - \eta_\ell \mathcal{E}_\ell)^2,
\end{aligned}
$$

where we reuse this transfer of $A_\ell^\top$ from the right to the left, and then use the definition of $\mathcal{E}_\ell$. □

Note from the above expression that for any $\ell' > \ell$, $\mathcal{E}_{\ell'}$ is a polynomial in $\mathcal{E}_\ell$. As a result, $\mathcal{E}_\ell$ and $\mathcal{E}_{\ell'}$ commute for all pairs $(\ell, \ell')$.

***Telescoping the Error.*** For the sake of conciseness in the further expressions, we further introduce the following notation.

**Definition 3.** Define $M_\ell := \prod_{l<\ell}(I - \eta_l \mathcal{E}_l)^2$ and $N_\ell^\top := \prod_{l<\ell}(I - \gamma_l \mathcal{E}_l)$, where we use the convention that

$$
\prod_{l<\ell} U_l = U_1 U_2 \cdots U_{\ell-1},
$$

and $M_0 = N_0 = I$.

By Lemma 3, we have $\mathcal{E}_\ell = \mathcal{E}_0 M_\ell$ and $\mathcal{V}_\ell = \mathcal{V}_0 M_\ell$. Further, $C_\ell = N_\ell C_0$.[3]

This leads to the following estimate of the value of $D_\ell$.

**Lemma 4.** *Let $W_*$ be the Nyström regression parameter for the data $(A, B)$. For any $\ell \geq 0$, we have*

$$
D_\ell = W_*(I - N_\ell) C_0.
$$

*Proof.* Since $D_0 = 0$, we have

$$
D_\ell = \sum_{l<\ell} \gamma_l \mathcal{V}_l C_l = \mathcal{V}_0 \left( \sum_{l<\ell} \gamma_l M_l N_l \right) C_0
$$

$$
= W_* \mathcal{E}_0 \left( \sum_{l<\ell} \gamma_l M_l N_l \right) C_0 = W_* \left( \sum_{l<\ell} \gamma_l \mathcal{E}_0 M_l N_l \right) C_0
$$

$$
= W_* \left( \sum_{l<\ell} \gamma_l \mathcal{E}_l N_l \right) C_0,
$$

where we used the definition of $M_\ell$. But now, using the definition of $N_\ell$, notice that for any $l$,

$$
N_{l+1}^\top = N_l^\top (I - \gamma_l \mathcal{E}_l) \iff N_{l+1} = N_l - \gamma_l \mathcal{E}_l N_l \iff \gamma_l \mathcal{E}_l N_l = N_l - N_{l+1}.
$$

Consequently, we can write

$$
D_\ell = W_* \left( \sum_{l<\ell} (N_l - N_{l+1}) \right) C_0 = W_*(I - N_\ell) C_0. \qquad \square
$$

---

[3]Note that the order of multiplication is flipped here, which follows notationally since we defined the transpose $N_\ell^\top$ above. Of course, all the $\mathcal{E}_l$ commute, so this is not very important in this case, but this subtle distinction will matter more in subsequent analysis.

Recall that $\hat{D}_* = W_* C_0$, meaning that the error $D_\ell - \hat{D}_*$ is $W_* N_\ell C_0$. Thus, this lemma captures a basic fact: as the size of the matrix $N_\ell$ decays, the output $D_\ell$ gets closer to the target output $\hat{D}_*$. Thus, our main focus is to characterize the decay of this matrix. We shall show this by arguing that the matrices $\mathcal{E}_\ell$ quickly become well-conditioned through the course of the iterations, and consequently $N_\ell$ decays quickly.

Before proceeding, note that in general, $\mathcal{E}_0$ may have zero eigenvalues, which are left unchanged by the main dynamics (see below), meaning that $I - N_\ell$ may even asymptotically have a large eigenvalue corresponding to vectors in the kernel of $\mathcal{E}_0$. However, we observe that $W_* = \mathcal{V}_0 \mathcal{E}_0^\dagger$, and thus any energy in $C_0$ that lies in the kernel of $\mathcal{E}_0$ is irrelevant to the prediction $\hat{D}_*$. As such, it is equivalent for us to analyse the two-norm of the matrix $\tilde{N}_\ell := (I - P_0) N_\ell$, where $P_0$ projects onto the kernel of $\mathcal{E}_0$. Instead of this notational complication, we will henceforth just assume that $\mathcal{E}_0$ is full rank (i.e., all of its eigenvalues are nonzero), and mention where changes need to be made to handle the general case.

***Conditioning of $\mathcal{E}_\ell$.*** Let us then begin with some notation: for a positive (semi-)definite symmetric matrix $\mathcal{M}$, we let $\lambda^i(\mathcal{M})$ denote its $i$th largest eigenvalue, $\overline{\lambda}(\mathcal{M})$ denote its largest eigenvalue, and $\underline{\lambda}(\mathcal{M})$ denote its smallest (nonzero) eigenvalue. Notice that $\overline{\lambda}(\cdot) = \lambda^1(\cdot)$, and $\underline{\lambda}(\mathcal{M}) = \lambda^r(\mathcal{M})$, where $r$ is the rank of $\mathcal{M}$ (we will work with full rank $\mathcal{M}$, but be cognizant of rank sensitivity). Let $v^i(\mathcal{M})$ denote the corresponding eigenvector. We further introduce the notation

$$\lambda_\ell^i = \lambda^i(\mathcal{E}_\ell), \quad \overline{\lambda}_\ell = \overline{\lambda}(\mathcal{E}_\ell), \quad \textit{and} \quad \underline{\lambda}_\ell = \underline{\lambda}(\mathcal{E}_\ell).$$

Our first observation is that the iterations leave the eigenstructure of $\mathcal{E}_\ell$ undisturbed.

**Lemma 5.** *Suppose that $v$ is an eigenvector of $\mathcal{E}_0$. Then it is also an eigenvector of $\mathcal{E}_\ell$ for any $\ell > 0$. Further, if $v \in \ker(\mathcal{E}_0)$ then $v \in \ker(\mathcal{E}_\ell)$.*

*Proof.* We have $\mathcal{E}_0^n v = \lambda^n v$. But note that $\mathcal{E}_\ell$ is some polynomial in $\mathcal{E}_0$, i.e., for some finite number of coefficients $\alpha_n$, $\mathcal{E}_\ell v = \sum_{n \geq 0} \alpha_n \mathcal{E}_0^n v = (\sum_{n \geq 0} \alpha_n \lambda^n) v$, and the claim follows. Crucially, observe that since $\mathcal{E}_\ell = M_\ell \mathcal{E}_0$, we have $\alpha_0 = 0$, and so if $\mathcal{E}_0 v = 0$, then $\mathcal{E}_\ell v = 0$ as well. □

Since the eigenstructure of $\mathcal{E}_0$ is preserved, we can begin to study the behaviour of its eigenvalues in a simplified way. The critical relationship for our analysis is the following iterative structure on the largest and smallest (nonzero) eigenvalues of $\mathcal{E}_\ell$.

**Lemma 6.** *Suppose that $\forall \ell, \eta_\ell \leq \overline{\lambda}_\ell^{-1}/3$. Then for all $\ell$,*

$$\overline{\lambda}_{\ell+1} = (1 - \eta_\ell \overline{\lambda}_\ell)^2 \overline{\lambda}_\ell \quad \textit{and} \quad \underline{\lambda}_{\ell+1} = (1 - \eta_\ell \underline{\lambda}_\ell)^2 \underline{\lambda}_\ell.$$

*Proof.* Suppose $u^1, u^2$ are two eigenvectors of $\mathcal{E}_0$, with positive eigenvalues $\mu_0, \nu_0$. Since $u^1, u^2$ are also eigenvectors of $\mathcal{E}_\ell$, denote the eigenvalues for the latter as $\mu_\ell, \nu_\ell$. Note that these remain nonnegative. Indeed, by multiplying the dynamics for $\mathcal{E}_{\ell+1}$ by, say, $u^1$, we have

$$\mu_{\ell+1} u^1 = \mathcal{E}_{\ell+1} u^1 = \mathcal{E}_\ell (I - \eta_\ell \mathcal{E}_\ell)^2 u^1 = (1 - \eta_\ell \mu_\ell)^2 \mu_\ell,$$

and similarly for $\nu_\ell$. We will first inductively show that if $\mu_0 > \nu_0 \iff \mu_\ell > \nu_\ell$ for all $\ell$. In other words, not only are the eigenvectors of $\mathcal{E}_0$ stable under the iterations, but also the ordering of the eigenvalues.

We thus note that the claim follows directly from this result. Indeed, let $v^1$ be the eigenvector corresponding to $\overline{\lambda}_0$. Then we see that it remains eigenvector corresponding to $\overline{\lambda}_\ell$ for all $\ell$. But then using our observation above with $u^1 = v^1$, we get the result for $\overline{\lambda}_{\ell+1}$. A similar argument works for $\underline{\lambda}_{\ell+1}$. Note that nothing per se demands that these eigenvalues have unit multiplicity, and the argument is completely insensitive to this.

To see the ordering claim on $\mu_\ell, \nu_\ell$, then, we observe that

$$\begin{aligned} \mu_{\ell+1} - \nu_{\ell+1} &= (1 - \eta_\ell \mu_\ell)^2 \mu_\ell - (1 - \eta_\ell \nu_\ell)^2 \nu_\ell \\ &= (\mu_\ell - \nu_\ell) - 2\eta(\mu_\ell^2 - \nu_\ell^2) + \eta^2(\mu_\ell^3 - \nu_\ell^3) \\ &= (\mu_\ell - \nu_\ell)\left(1 - 2\eta_\ell(\mu_\ell + \nu_\ell) + \eta_\ell^2(\mu_\ell^2 + \nu_\ell^2 + \mu_\ell \nu_\ell)\right). \end{aligned}$$

Now, if the term multiplying $(\mu_\ell - \nu_\ell)$ is nonnegative, then the claim follows. To see when this occurs, let us set $\eta_\ell = \eta \overline{\lambda}_\ell^{-1}$, and pull $\overline{\lambda}^{-1}$ within the brackets. We are left with an expression of the form

$$(1 - 2\eta(x + y) + \eta^2(x^2 + y^2 + xy)),$$

where $(x, y) \in (0, 1)$. For $\eta \leq 2/3$, the minimum over this range occurs when $x = y = 1$, and takes the value

$$1 - 4\eta + 3\eta^2.$$

It is a triviality to show that this function is nonnegative for $\eta \leq 1/3$, and so we are done. $\qquad\square$

With this in hand, we will argue that $\underline{\lambda}_\ell \to \overline{\lambda}_\ell$ at a quadratic convergence rate after an initial burn-in period. Of course, this is equivalent to saying that $\kappa_\ell := \overline{\lambda}_\ell/\underline{\lambda}_\ell \to 1$. Notice that this $\kappa_\ell$ is precisely the condition number of $\mathcal{E}_\ell$ (which in turn is the square of the condition number of $A_\ell$ mentioned in our discussion in §5).

**Lemma 7.** *Define $\kappa_\ell = \overline{\lambda}_\ell/\underline{\lambda}_\ell$, and $\theta_\ell = \kappa_\ell - 1$. If $\eta_\ell \leq \overline{\lambda}^{-1}/3$, then $\theta_{\ell+1} \leq \theta_\ell \exp(-5\eta_\ell\overline{\lambda}_\ell/3)$. Further, if $\eta_\ell\overline{\lambda}_\ell = 1/3$, then $\theta_{\ell+1} \leq 3\theta_\ell^2/4$.*

*Proof.* For the sake of succinctness, let us write $\eta_\ell = \eta, \overline{\lambda}_\ell = \overline{\lambda}, \underline{\lambda}_\ell = \underline{\lambda}, \theta_\ell = \theta$, and $\theta_{\ell+1} = \theta_+$. Then, using the previous lemma, we can write

$$\kappa_{\ell+1} = \theta_+ + 1 = (\theta + 1)\left(\frac{1 - \eta\overline{\lambda}}{1 - \eta\underline{\lambda}}\right)^2.$$

But notice that

$$\frac{1 - \eta\overline{\lambda}}{1 - \eta\underline{\lambda}} = 1 + \eta\frac{\overline{\lambda} - \underline{\lambda}}{1 - \eta\underline{\lambda}} = 1 - \eta\frac{\overline{\lambda}(1 - 1/(\theta + 1))}{1 - \eta\overline{\lambda}/(\theta + 1)} = 1 - \frac{\theta\eta\overline{\lambda}}{\theta + 1 - \eta\overline{\lambda}}.$$

Thus,

$$\theta_+ + 1 = (\theta + 1)\left(1 - 2\frac{\theta\eta\overline{\lambda}}{(\theta + 1 - \eta\overline{\lambda})} + \frac{\theta^2\eta^2\overline{\lambda}^2}{(\theta + 1 - \eta\overline{\lambda})^2}\right)$$

$$= \theta + 1 - \theta\frac{\eta\overline{\lambda}}{1 - \eta\underline{\lambda}}\left(2 - \frac{\theta\eta\overline{\lambda}}{(\theta + 1 - \eta\overline{\lambda})}\right).$$

But the bracketed term is nonincreasing in $\eta\overline{\lambda}$ (in the negative term, the numerator increases and the denominator decreases with this value), and so

$$2 - \frac{\theta\eta\overline{\lambda}}{\theta + 1 - \eta\overline{\lambda}} \geq 2 - \frac{\theta}{3\theta + 3 - 1} = \frac{5\theta + 4}{3\theta + 2} \geq \frac{5}{3}.$$

Thus, we have

$$\theta_{\ell+1} \leq \theta_\ell - \theta_\ell \cdot \frac{5/3\eta_\ell\overline{\lambda}_\ell}{1 - \eta_\ell\underline{\lambda}_\ell} \implies \theta_{\ell+1} \leq \theta_\ell \exp\left(-\frac{5\eta_\ell\overline{\lambda}_\ell/3}{1 - \eta_\ell\underline{\lambda}_\ell}\right) \leq \theta_\ell \exp(-\eta_\ell\overline{\lambda}_\ell).$$

On the other hand (going back to the notation that suppresses $\ell$), if we set $\eta\overline{\lambda} = L$, then since

$$\frac{1 - \eta\overline{\lambda}}{1 - \eta\underline{\lambda}} = \frac{1 - L}{(\theta + 1 - L)/(\theta + 1)},$$

we find by doing the long multiplication that

$$(\theta_+ + 1) = \frac{(\theta + 1)^3(1 - L)^2}{(\theta + 1 - L)^2}$$

$$\iff \theta_+ = \frac{\theta(1 - 4L + 3L^2) + \theta^2(2 - 6L + 3L^2) + \theta^3(1 - L)^2}{(\theta + 1 - L)^2}.$$

Setting $L = \eta_\ell\overline{\lambda}_\ell = 1/3$, the linear term in $\theta$ vanishes, and we end up with

$$\theta_+ = \frac{\theta^2/3 + 4\theta^3/9}{(\theta + 2/3)^2} = \theta^2 \cdot \frac{4\theta + 3}{(3\theta + 2)^2} \leq \frac{3\theta^2}{4}. \qquad\square$$

The statement of this Lemma has shown two decay modes of $\theta_\ell = (\kappa_\ell - 1)$. Firstly, as long as $\eta_\ell \leq \overline{\lambda}_\ell^{-1}/3$, this quantity decays at least at a linear rate. Further, if $\eta_\ell$ is set to exactly $\overline{\lambda}_\ell^{-1}/3$, then once $\theta_\ell$ dips below 1, which occurs after about $3\log(\theta_0)/5$ iterations, we can exploit the quadratic bound in $\theta$ to recover a stronger convergence rate. Together, this yields the following convergence behaviour for $\kappa_\ell$.

**Lemma 8.** *Let* $\kappa_\ell := \overline{\lambda}_\ell/\underline{\lambda}_\ell$. *If for all* $\ell$, $\eta_\ell\overline{\lambda} = 1/3$, *then* $\kappa_\ell - 1 \leq \varepsilon$ *for all*

$$\ell \geq L = 2 + \lceil (3/5)\log(\kappa_0)\rceil + \lceil\log_2(\log(4/3\varepsilon))\rceil.$$

*Proof.* First using the geometric decay in Lemma 7, we know that $\theta_\ell \leq (\kappa_0 - 1)\exp(-5(\ell-1)/3)$. Set $\ell_0 = 2 + 3/5\log(\kappa_0 - 1)$. Then for $\ell \geq \ell_0$, we have $\theta_{\ell_0} \leq 1/e$. Further, for iterations beyond this $\ell_0$, the supergeometric decay $\theta_{\ell+1} \leq 3\theta_\ell^2/4$ in Lemma 7 implies that

$$3\theta_{\ell+1}/4 \leq (3\theta_\ell/4)^2 \iff T_{\ell+1} \leq 2T_\ell,$$

where $T_\ell := \log(3\theta_\ell/4)$. Thus,

$$T_\ell \leq 2^{\ell-\ell_0}T_{\ell_0} \leq -2^{\ell-\ell_0} \implies \theta_\ell \leq 4/3\exp(-2^{\ell-\ell_0}).$$

Setting this $< \varepsilon$ gives the claim. $\qquad\square$

We again note that this $\kappa_\ell$ is the square of the condition number of $A_\ell$ - however, this distinction is easily accommodated due to the logarithmic dependence on it - the only change is that the $3/5$ above increases to $6/5$.

***Back to*** $N_\ell$. The conditioning of $\mathcal{E}_\ell$ yields a direct control on the behaviour of $N_\ell$, as encapsulated below. We note that if $\mathcal{E}_0$ is not full rank, the statement holds for $\tilde{N}_\ell = (I - P_0)N_\ell$, where $P_0$ projects onto the kernel of $\mathcal{E}_0$.

**Lemma 9.** *For all* $\ell$, $\|N_\ell\|_2 \leq \exp(-\sum_{l<\ell}\gamma_l\underline{\lambda}_l)$. *Further, if* $\forall \ell, \eta_\ell = \overline{\lambda}^{-1}/3$ *and* $\gamma_\ell = \overline{\lambda}^{-1}$, *then* $\|N_\ell\|_2 \leq \varepsilon$ *for all* $\ell \geq L + 1$, *where*

$$L = 2 + \lceil 3/5\log(\kappa_0)\rceil + \lceil\log_2(\log 4/3\varepsilon)\rceil.$$

*Proof.* Since $N_{l+1} = (I - \gamma_l\mathcal{E}_l)N_l$, we immediately have

$$\|N_\ell\|_2 \leq \prod_{l<\ell}\|I - \gamma_l\mathcal{E}_l\|_2.$$

But note that $\|I - \gamma_l\mathcal{E}_l\|_2 = (1 - \gamma_l\underline{\lambda}_l)$. Thus, we immediately have $\|N_\ell\| \leq \prod(1 - \gamma_l\underline{\lambda}_l) \leq \exp(\sum_{l<\ell}\gamma_l\underline{\lambda}_l)$. Further, recall from Lemma 8 that if $\eta_\ell = \overline{\lambda}^{-1}/3$ for all $\ell$, then $\underline{\lambda}_\ell \geq \overline{\lambda}_\ell/(1+\varepsilon)$ for all $\ell \geq L$. But then, if $\eta_\ell = \overline{\lambda}^{-1}$, we have

$$1 - \eta_\ell\underline{\lambda} \leq 1 - \underline{\lambda}/\overline{\lambda} \leq \varepsilon/(1+\varepsilon) \leq \varepsilon,$$

yielding the claimed bound on $\|N_\ell\|_2$. $\qquad\square$

***Finishing the Error Control.*** With all the pieces in place, we conclude the main argument.

*Proof of Theorem 1.* By Lemma 4, we have $D_\ell - \hat{D}_* = W_*N_\ell C_0$, and consequently,

$$\|D_\ell - \hat{D}_*\|_F \leq \|W_*N_\ell\|_F\|C_0\|_F \leq \sqrt{\mathrm{rank}(W_*N_\ell)}\|W_*N_\ell\|_2\|C_0\|_F.$$

But, since $W_*N_\ell \in \mathbb{R}^{d'\times d}$, the rank about is at most $d'$. Finally, $\|W_*N_\ell\|_2 \leq \|N_\ell\|_2\|W_*\|_2$. Thus, the claim follows as soon as $\|N_\ell\|_2 \leq \varepsilon/(\sqrt{d'}\|W_*\|_2\|C_0\|_F)$, for which we may invoke Lemma 9. $\qquad\square$

Again, recall that for the purposes of error, if $\mathcal{E}_0$ were not full rank, then we could instead replace $N_\ell$ by $\tilde{N}_\ell$ in all statements above, and thus the claim also extends to this situation.

***Comment on Rates $\eta, \gamma$.*** Going back to the notation of Algorithm 1, we reparametrise $\gamma_\ell = \rho_\ell = \overline{\lambda}_\ell^{-1} = \|A_\ell\|_2^{-2}$, and $\eta_\ell = \rho_\ell/3$. It is worth discussing briefly how we may go about estimating this $\rho_\ell$.

A simple observation in this setting is that if we assume that we know $\overline{\lambda}_0 = \|\mathcal{E}_0\|_0 = \|A\|_2^2$ to begin with, then it is a simple matter to compute the subsequent values of $\overline{\lambda}_\ell$, since if we set $\eta_\ell, \gamma_\ell$ as per the above, this is directly computed iteratively via

$$\overline{\lambda}_{\ell+1} = \overline{\lambda}_\ell (1 - 1/3)^2 = 4\overline{\lambda}_\ell/9.$$

In fact, under this assumption, we may avoid further numerical stability issues by first recaling $A$ to have 2-norm 1, and subsequently simply rescaling the matrices up $A_\ell, B_\ell$ by $3/2$ after each update to maintain the invariant $\overline{\lambda}_\ell = \overline{\lambda}_0$—the behaviour of $N_\ell$ remains the same, and convergence rates are only driven by the conditioning of $A_\ell$. So, equivalently, the question we must concern ourselves with is finding $\|A\|_2$. Of course, this is quite cheap, since we can compute this simply by power iteration, which involves only matrix-vector products rather than matrix-matrix products. Nevertheless, note that power iteration is only linearly convergent (i.e., the number of iterations needed to find a $\varepsilon$-approximation of the top eigenvalue is $\Theta(\log(1/\varepsilon))$, so in full generality, this procedure would destroy the second order convergence.

In our simulations of §5, we indeed implemented these iterations by carrying out power iteration. Note that practically, then, the method essentially retains its second order behaviour despite this approximation. One aspect of this lies in the fact that even if we underestimate the top eigenvalue by a small amount, and so undertune $\eta, \gamma$ by a slight amount, the second order bound of Lemma 8 decays gracefully, and so retains practical resilience. Of course, characterising exactly how loose this can be requires more precise analysis, which we leave for future work.

## F.2 Distributed Analysis

Beginning again with Algorithm 1, with $S = I_n, M > 1$, we need to analyse the iterations

$$\forall \mu, A_{\ell+1}^\mu = A_\ell^\mu (I - \eta_\ell^\mu A_\ell^{\mu,\top} A_\ell^\mu), \tag{7}$$
$$\forall \mu, B_{\ell+1}^\mu = B_\ell^\mu (I - \eta_\ell^\mu A_\ell^{\mu,\top} A_\ell^\mu),$$
$$C_{\ell+1} = C_\ell - \frac{1}{m} \sum_\mu \gamma_\ell A_\ell^\mu A_\ell^{\mu,\top} C_\ell,$$
$$D_{\ell+1} = D_\ell + \frac{1}{m} \sum_\mu \gamma_\ell B_\ell^\mu A_\ell^{\mu,\top} C_\ell.$$

We note that here we have set $\gamma_\ell$ to be constant across all machines, and nominally it is not pegged to $\|A_\ell^\mu\|_2^{-2}$, which could vary across machines. However, we will analyze this method under the normalization assumption that

$$\forall \mu, \|A_0^\mu\|_2 = 1.$$

In this setting, we will show that the choices $\eta_\ell^\mu = \|A_\ell^\mu\|_2^{-2}/3$ are also the same for each machine, and then set $\gamma_\ell = 3\eta_\ell^\mu$ (which in turn is also the same for every machine). Thus, in the setting we analyze, the iterations above are faithful to the structure of Algorithm 1.

Note, of course, that this distributed data computes the Nyström approximation of

$$\begin{bmatrix} A & C \\ B & D \end{bmatrix},$$

where $C, D$ are as in the iteration, while

$$A = \begin{bmatrix} A^1 & A^2 & \cdots & A^m \end{bmatrix}, B = \begin{bmatrix} B^1 & B^2 & \cdots & B^m \end{bmatrix}.$$

We will work in the noise-free regime, wherein there exists a matrix $W_*$ such that

$$\begin{bmatrix} B & D \end{bmatrix} = W_* \begin{bmatrix} A & C \end{bmatrix},$$

and $C \in$ column-span$(A)$. Of course, this $W_*$ also equals the Nyström parameter for this matrix (§B.1). The second condition ensures that the rank of this matrix is the same as that of $A$, and is needed to ensure that we can actually infer $W_*$ in the directions constituted by the columns of $C$ (without which one cannot recover $D$).

***Per-machine and Global Signal Energies.*** We begin with defining the energy and correlation matrices in analogy to the previous section. As before, $A_0^\mu = A^\mu, B_0^\mu = B^\mu, C_0 = C, D_0 = 0$.

**Definition 4.** We define the per-machine objects

$$\mathcal{E}_\ell^\mu := A_\ell^\mu A_\ell^{\mu,\top} \text{ and } \mathcal{V}_\ell^\mu := B_\ell^\mu A_\ell^{\mu,\top},$$

and the per-machine values

$$\overline{\lambda}_\ell^\mu = \lambda^1(\mathcal{E}_\ell^\mu), \underline{\lambda}_\ell^\mu = \lambda^{r(\mu)}(\mathcal{E}_\ell^\mu),$$

where $r(\mu)$ is the rank of $\mathcal{E}_0^\mu$. Further, we define the global signal energy

$$\overline{\mathcal{E}}_\ell := \frac{1}{m} \sum \mathcal{E}_\ell^\mu.$$

Note that by our assumption, $\|\mathcal{E}_0^\mu\|_2 = 1$ for all $\mu$. Further, the behaviour of $\mathcal{E}_\ell^\mu, \mathcal{V}_\ell^\mu$ is identical to that in the centralised case, i.e.,

**Lemma 10.** *For all $\mu, \ell$,*

$$\mathcal{E}_{\ell+1}^\mu = \mathcal{E}_\ell^\mu (I - \eta_\ell^\mu \mathcal{E}_\ell^\mu)^2 \text{ and } \mathcal{V}_{\ell+1}^\mu = \mathcal{V}_\ell^\mu (I - \eta_\ell^\mu \mathcal{E}_\ell^\mu)^2.$$

*As a consequence, for every $\ell, \mu$, it holds that*

$$\mathcal{V}_\ell^\mu = W_* \mathcal{E}_\ell^\mu.$$

*Finally, for any $\ell$,*

$$C_{\ell+1} = (I - \gamma_\ell \overline{\mathcal{E}}_\ell) C_\ell$$

*Proof.* The iterations for $\mathcal{E}_{\ell+1}^\mu, \mathcal{V}_{\ell+1}^\mu$ follow identically to the proof of Lemma 3. For the second claim, notice that we have

$$B^\mu = W_* A^\mu \implies \mathcal{V}_0^\mu = W_* \mathcal{E}_0^\mu$$

by multiplying by $A^{\mu,\top}$ on both sides. Then the claimed invariance follows inductively.

Finally, noting that $\mathcal{E}_\ell^\mu = A_\ell^\mu A_\ell^{\mu,\top}$, we immediately have

$$C_{\ell+1} = \left(I - \gamma_\ell \cdot \frac{1}{m} \sum_\mu \mathcal{E}_\ell^\mu\right) C_\ell = (I - \gamma_\ell \overline{\mathcal{E}}_\ell) C_\ell \qquad \square$$

In analogy with the centralized case, this leads us to define the following matrices.

**Definition 5.** We define

$$M_\ell^\mu = \prod_{l < \ell} (I - \eta_\ell^\mu \mathcal{E}_\ell^\mu)^2 \text{ and } N_\ell^\top := \prod_{l < \ell} (I - \gamma_\ell \overline{\mathcal{E}}_\ell),$$

where again $\prod$ is interpreted as multiplication from the right, and $M_0^\mu = N_0 = I_d$.

Succinctly, then, we can write $\mathcal{E}_\ell^\mu = \mathcal{E}_0^\mu M_\ell^\mu$ and $C_\ell = N_\ell C_0$.

***Telescoping the Error.*** This notation allows us to set up the following analogue of Lemma 4.

**Lemma 11.** *Under the distributed iterations without noise, it holds for all $\ell$ that*

$$D_\ell = W_* (I - N_\ell) C_0.$$

*Proof.* Observe that

$$D_\ell = \sum_{l < \ell} \gamma_l \cdot \sum_\mu \frac{\mathcal{V}_l^\mu}{m} C_l.$$

Now, $\mathcal{V}_l^\mu = W_* \mathcal{E}_l^\mu$, and so

$$\frac{1}{m} \sum_\mu \mathcal{V}_l^\mu = W_* \cdot \frac{1}{m} \sum_\mu \mathcal{E}_l^\mu = W_* \overline{\mathcal{E}}_l.$$

Thus, we have

$$D_\ell = W_* \left( \sum_{l < \ell} \gamma_l \overline{\mathcal{E}}_l N_l \right) C_0.$$

But

$$N_{l+1} = (I - \gamma_l \mathcal{E}_l) N_l \iff \gamma_l \overline{\mathcal{E}}_l N_l = N_l - N_{l+1},$$

and so the term in the brackets telescopes to $N_0 - N_\ell = I - N_\ell$. $\qquad\square$

***Conditioning.*** Of course, again, $D = W_* C = W_*(I)C_0$. So, to gain error control, we only need to argue (as before) that $\|N_\ell\|_2$ vanishes quickly with $\ell$. As before, this relies strongly on the condition number of $\overline{\mathcal{E}}_\ell$. We note, again, that we will simply assume that $\overline{\mathcal{E}}_0$ is full-rank, since rank-deficiency is rendered moot in this case by the fact that $C$ lies in the column span of $A$ (and hence, is orthogonal to the kernel of $\overline{\mathcal{E}}_0$). However, the individual $\mathcal{E}_0^\mu$ may not be full rank, and so the distinction between $\underline{\lambda}(\mathcal{E}^\mu)$ and the smallest eigenvalue of $\mathcal{E}^\mu$ (which is usually 0) should not be forgotten, although it will not matter very much for our expressions.

To begin with, we note that since the per-machine $A^\mu$, $B^\mu$ iterations are identical to the central case, the corresponding $\mathcal{E}^\mu$ are conditioned at the same quadratic rate we saw previously. We formally state this below.

**Lemma 12.** *For all $\ell$, set $\eta_\ell^\mu := (\overline{\lambda}_\ell^\mu)^{-1}/3$, and define $\kappa_\ell^\mu = \overline{\lambda}_\ell^\mu / \underline{\lambda}_\ell^\mu$.*

*If $\ell \geq L(\varepsilon) := 2 + \lceil 3/5 \log(\max_\mu \kappa_0^\mu) \rceil + \lceil \log_2(\log(4/3\varepsilon)) \rceil$, then $\max_\mu \kappa_\ell^\mu \leq 1 + \varepsilon$.*

*Proof.* Apply Lemma 8. $\qquad\square$

Of course, $\overline{\mathcal{E}}_\ell$, as an average, will not have the same conditioning behaviour. In fact, this is strongly sensitive to the diversity index $\alpha$ from Definition 1, as captured below.

**Lemma 13.** *Suppose that $\lambda_0^\mu$ is constant across machines, and for all $\mu, \ell, \eta_\ell^\mu = (\overline{\lambda}_\ell^\mu)^{-1}/3$. Define*

$$\overline{\kappa}_\ell = \overline{\lambda}(\overline{\mathcal{E}}_\ell)/\underline{\lambda}(\overline{\mathcal{E}}_\ell).$$

*If $\ell \geq L(\varepsilon)$, then $\overline{\kappa}_\ell \leq \dfrac{1 + \varepsilon}{\alpha}$,*

*where $L(\varepsilon)$ is the expression in Lemma 12. Further, for the same range of $\ell$, for all $\mu$,*

$$\underline{\lambda}(\overline{\mathcal{E}}_\ell) \geq \frac{\overline{\lambda}_\ell^\mu \cdot \alpha}{1 + \varepsilon}.$$

*Proof.* Firstly, by Weyl's inequality, notice that

$$\overline{\lambda}(\overline{\mathcal{E}}_\ell) \leq \frac{1}{m} \sum_\mu \overline{\lambda}(\mathcal{E}_\ell^\mu).$$

Further, recall that $\overline{\lambda}(\mathcal{E}_0^\mu) = 1$ for all $\mu$, and we set $\eta_\ell^\mu = (\overline{\lambda}_\ell^\mu)^{-1}/3$. Thus, each of these $\overline{\lambda}_\ell^\mu$s are infact identical (and equal to $(4/9)^\ell$). Thus,

$$\forall \mu, \overline{\lambda}(\overline{\mathcal{E}}_\ell) \leq \overline{\lambda}_\ell^\mu.$$

Now, let $P^\mu$ be the projection onto the column space of $A^\mu$. Then we note that for any vector $v$,

$$v^\top \mathcal{E}_\ell^\mu v = (P^\mu v)^\top \mathcal{E}_\ell^\mu (P^\mu v).$$

Indeed, any component in $v$ orthogonal to the column space must lie in the kernel of $\mathcal{E}_0^\mu = A^\mu A^{\mu,\top}$, and we know that this kernel is invariant across the iterations. Further, note that since $P^\mu v$ lies in this

column space, it is orthogonal to any eigenvector of $\mathcal{E}_\ell^\mu$ with zero eigenvalue, and so we can then conclude that

$$(P^\mu v)^\top \mathcal{E}_\ell^\mu (P^\mu v) \geq \underline{\lambda}_\ell^\mu \|P^\mu v\|_2^2.$$

But then, if $\ell \geq L(\varepsilon)$, then for any unit vector $v$, we have (for any $\mu$ in the last inequalities) that

$$v^\top \overline{\mathcal{E}}_\ell v = \frac{1}{m} \sum_\mu v^\top \mathcal{E}_\ell^\mu v$$

$$\geq \frac{1}{m} \sum_\mu \underline{\lambda}_\ell^\mu \|P^\mu v\|_2^2 = \frac{1}{m} \sum_\mu \frac{\overline{\lambda}_\ell^\mu}{\kappa_\ell^\mu} \|P^\mu v\|_2^2$$

$$\geq \frac{\overline{\lambda}_\ell^\mu}{1+\varepsilon} \frac{1}{m} \sum_\mu \|P^\mu v\|_2^2$$

$$\geq \frac{\overline{\lambda}_\ell^\mu}{1+\varepsilon} \cdot \alpha,$$

where we used the equality of the $\overline{\lambda}_\ell^\mu$, the fact that $\kappa_\ell^\mu \leq 1+\varepsilon$, and finally the definition of $\alpha$. Since $v$ is a unit vector, we conclude that any Rayleigh quotient of $\overline{\mathcal{E}}_\ell$ is so lower bound, ergo

$$\forall \mu, \underline{\lambda}(\overline{\mathcal{E}}_\ell) \geq \frac{\overline{\lambda}_\ell^\mu}{1+\varepsilon} \cdot \alpha.$$

Putting this together with the upper bound on $\overline{\lambda}(\overline{\mathcal{E}}_\ell)$, we immediately conclude that

$$\overline{\kappa}_\ell \leq \frac{1+\varepsilon}{\alpha}. \qquad \square$$

At a high level, $\alpha^{-1}$ is the limiting condition number of the $\overline{\mathcal{E}}_\ell$s, where deviation from perfect conditioning occurs only due to how the various $\mathcal{E}_\ell^\mu$ energise distinct subspaces for large $\mu$. We note that this dependence is tight—if all the $\mathcal{E}^\mu$ share the top eigenvector, then the upper bound on $\overline{\lambda}(\overline{\mathcal{E}}_\ell)$ is exact, while the lowest nonzero eigenvector of $\overline{\mathcal{E}}_\ell$ is precisely the direction that achieves the minimum in the definition of the diversity index.

***Concluding the argument.*** With this in hand, we can argue the decay of $N_\ell$, and so attain error control.

*Proof of Theorem 2.* Recall that $D - \hat{D}_\ell = W_* N_\ell C_0$. We again assume that $\overline{\lambda}_0^\mu = 1$ for all $\mu$, and that $\eta_\ell^\mu = (\overline{\lambda}_\ell^\mu)^{-1}/3$. We set $\gamma_\ell = (\overline{\lambda}_\ell^\mu)^{-1} \leq \overline{\lambda}(\overline{\mathcal{E}}_\ell)^{-1}$. Then, as before, we have (restricted to the appropriate subspace if $\overline{\mathcal{E}}_0$ is not full rank)

$$\|I - \eta_\ell \overline{\mathcal{E}}_\ell\|_2 \leq (1 - 1/\overline{\kappa}_\ell) \leq \exp(-1/\overline{\kappa}_\ell).$$

By Lemma 13, for $\ell \geq L(1)$, we have $\overline{\kappa}_\ell \leq 2/\alpha$, and so

$$\|N_\ell\| \leq \exp(-\sum_{l=L(1)}^\ell 1/\overline{\kappa}_\ell) \leq \exp(-\alpha(\ell - L(1))/2),$$

which is smaller than $\iota\varepsilon$ if

$$\ell \geq L(1) + \frac{2}{\alpha} \log \frac{1}{\iota\varepsilon}.$$

Of course, $L(1) = 3 + \lceil 3/5 \log(\max_\mu \kappa_0^\mu) \rceil$, and setting $\iota$ so small that error to $\varepsilon$ follows in the same way as the proof of Theorem 1 concludes the argument. $\qquad \square$

***Comment on Rates.*** We note that, in the distributed setting, within-machine computation is typically much cheaper than across-machine communication. As a result, the protocol we have analyzed above, wherein each machine begins with the same value of $\overline{\lambda}_0^\mu$, is cheap to follow by using power-iteration at each machine. Given this choice, the value $\overline{\lambda}_\ell^\mu$ is simply equal to $(4/9)^\ell$ for every machine, and setting the learning rate $\eta_\ell^\mu = \rho_\ell^\mu/3$ for $\rho_\ell^\mu = (9/4)^\ell$ is trivial, and does not require any communication.

We also note that setting $\gamma_\ell = \rho_\ell^\mu$ in each machine, and then averaging the results of the updates together is sufficient, since $\rho_\ell^\mu$ is constant across all machines. Thus, we recover exactly the structure presented in Algorithm 1, with $\gamma = 1, \eta = 1/3$. Note, again, that for the sake of stability, instead of working with decaying $A_\ell$ and exploding $\eta_\ell, \gamma_\ell$, we can again rescale each $A_\ell^\mu, B_\ell^\mu$ by $3/2$ after updates.

In general, if this normalization is not carried out, the machines may actually set their values for $\gamma_\ell$ as distinct $\gamma_\ell^\mu$s. The net effect would be that we need to analyze a slightly different version of $\overline{\mathcal{E}}_\ell$ that is sensitive to inter-machine variations in $\gamma_\ell^\mu$, which in turn would rely on how strongly $\overline{\lambda}_0^\mu$ varies across machines. We leave the study of such scenarios to future work.

