# OpenReview forum: "Linear Transformers Implicitly Discover Unified Numerical Algorithms"
_NeurIPS.cc/2025/Conference — NeurIPS 2025 poster_

### Official Review · Reviewer_fRWb · 2025-06-04

**Clarity:** 2
**Significance:** 2
**Originality:** 3
**Rating:** 4
**Confidence:** 2

**Summary:**

The paper train a linear transformer under 3 different regime (unconstrained, computed limited and distributed) on masked-block completion tasks showing that transformer is able to discover solution for the task under all regimes without any input on the task.

**Questions:**

- Which other tasks do you think can be interested to explore?
- Why do you choose linear transformer? How the method can be extended to non linear regime?
- Which real-world tasks do you think EAGLE can be implemented for?
- How stable numerically is the proposed algorithm? Do you found places where it diverge?
- Do you plan to open-source your code?

**Ethical Concerns:**

["NO or VERY MINOR ethics concerns only"]

**Final Justification:**

I believe the entire regime is limited, so I decided to keep my score.

**Limitations:**

The paper focus only on linear transformer and only in one task - which limited the contribution on real world scenarios.
I'll happy to hear which additional ideas the authors have to extend it to non linear regime and additional tasks.

**Paper Formatting Concerns:**

No formatting concerns

**Quality:**

2

**Strengths And Weaknesses:**

I think that the general idea of the paper is interesting and it is not a common area of research - I think it is a great bonus.
The paper regime is very limited and extension for additional models and regimes will be appreciated. It's unclear how transferable the EAGLE method is to real-world or non-linear tasks.

---

> ### Author Rebuttal · Authors · 2025-07-29
>
> We thank the reviewer for their positive feedback on our paper's core idea and for recognizing its novelty. We appreciate the thoughtful questions regarding the choice of a linear transformer and the path to extending our work. We are happy to provide additional details and context.
>
> ### On the Methodological Framework (Linear Transformer & Task Scope)
>
> The reviewer's main concern is that our focus on a single task with a linear transformer limits the contribution. We would like to clarify that these were deliberate methodological choices aimed at making the discovery of the transformer's implicit algorithm tractable and interpretable.
>
> - **The simple architecture was a result of principled simplification, not a starting assumption.** Our goal was to isolate the computational essence of the learned algorithm. To do this, we performed a systematic simplification process, as described in the **Algorithm extraction** section (p. 4) and **Appendix B.5** (p. 22). For example, we found that performance was maintained even after pruning the model from eight attention heads down to a single head per layer. The resulting "linear, attention-only" model is therefore not an arbitrary limitation, but rather the outcome of a principled approach to uncover the fundamental algorithm learned by the transformer.
>
> - **The transformer learns a simple, structured hyperparameter scheme.** A crucial part of algorithm discovery is identifying the hyperparameters. A surprising result of our work is that the transformer simplifies this process immensely. [cite_start]After the simplification procedure, we found that the learned weights revealed a clear, interpretable rule for the step-size parameters of EAGLE. As shown in **Section 4** ("Rationale for Weights", p. 5) and **Figure 5** (p. 7), the learned weights correspond to fixed constants ($\eta,\gamma$) combined with an automatic normalization based on the data's spectral norm ($\\|A\_\ell\\|\_{2}^{-2}$). This discovery transforms a potentially complex hyperparameter tuning problem into a simple, parameter-free update rule, providing strong evidence that the transformer is uncovering a robust and elegant numerical method.
>
> - **A "linear transformer" is not a linear model.** This is a critical point of clarification. A "linear transformer" is **not a linear function of its input data**. The attention update is fundamentally non-linear—specifically, it is **cubic in the data activations $Z$**. This is shown explicitly in the derivation in **Appendix C.3 (Eq. 4)**. The term 'linear' simply refers to the absence of an additional, point-wise non-linear activation (like a ReLU), which our simplification process found to be unnecessary for this task.
>
> ### On Generalizing the "EAGLE Method" and Future Work
>
> We agree with the reviewer that extending this work is an exciting direction. We believe reviewer comments may stem from a potential ambiguity in the term "EAGLE method," which can refer to either the specific algorithm or our discovery process.
>
> - **The `EAGLE` algorithm is task-specific.** The specific two-line update rule we call EAGLE is the algorithm the transformer discovered for the Nyström-type matrix completion problem. It is a powerful second-order solver for this task, but we would not expect it to solve a different problem like sparse regression.
>
> - **The discovery *process* is highly general.** The core transferable contribution of our work is the **methodology for discovering the implicit algorithm**. We demonstrate this concretely in **Appendix D** (p. 25), where we apply this same process to a related but distinct **parameter estimation** task, revealing a slightly modified, highly efficient version of EAGLE. This shows the process is not a one-off trick. We are actively applying this framework to other domains like sparse regression and high-dimensional classification.
>
> ### Answers to Specific Questions
>
> - **Why a linear transformer? How to extend to the non-linear regime?**
> As explained above, the choice was a result of a principled simplification to enable algorithm discovery. The model is already non-linear in its data dependence. Our ongoing work on other tasks (e.g., classification) shows that this same discovery process reveals algorithms where non-linear activation functions (like ReLU or GeLU) are kept, as they are essential for those tasks.
>
> - **What real-world tasks can EAGLE be implemented for?**
> The specific `EAGLE` algorithm is a second-order solver for least-squares problems based on Nyström approximation. It is therefore well-suited for real-world scenarios that rely on this technique, such as large-scale kernel methods, recommender systems, and statistical inference.
>
> - **How numerically stable is the algorithm? Do you find places where it diverges?**
> Practically, our method is stable. As we mention in the paper (lines 87, 188, 248 and appx C), the inner loop of EAGLE is closely related to the classical Newton-Schulz method for positive definite matrices. As such, we expect numerical stability similar to Newton-Schulz. Theoretically, this can get problematic when condition numbers start approaching 1/floating-point-precision, but empirically we expect stable behavior. This is demonstrated well in the Sparse-Suite Matrix Completion experiments (see response to reviewer rz5S), where we see very good performance.
>
> - **Do you plan to open-source your code?**
> Yes, we plan to release our code upon publication to encourage further research in this area.

---

> > ### Comment · Reviewer_fRWb · 2025-08-03
> > **Response to rebuttal**
> >
> > I appreciate the author's response.
> > All my concerns were addressed.
> > I still believe the entire regime is limited, so I decided to keep my score.

---

### Official Review · Reviewer_rz5S · 2025-07-03

**Clarity:** 3
**Significance:** 2
**Originality:** 3
**Rating:** 4
**Confidence:** 2

**Summary:**

The authors explored learning numerical algorithms for low-rank matrix completion with Transformers. The authors train transformers, without explicit algorithmic guidance, on tasks encompassing prediction, estimation, and Nyström extrapolation, imposing three computational regimes (centralized, computation-limited, and distributed) through architectural constraints. Analysis of the trained weights reveals a consistent, parameter-free, two-line iterative update rule applicable across all regimes.  EAGLE theoretically and empirically matches or surpasses classical solvers, automatically adapts to resource limitations, and demonstrates strong convergence properties.  The results highlight the potential for in-context algorithmic learning within transformers.

**Questions:**

- How does EAGLE's performance scale with increasing matrix size, different data distributions (e.g., sparsity, noise levels), and matrix structures?  What are its limitations in high-dimensional or highly structured scenarios?
- Given the inherent parallelism in matrix operations, how does EAGLE's performance compare to optimized parallel implementations of classical solvers on multi-core systems?  What is the potential for further parallelization within EAGLE itself?
- Is there any chance to evaluate EAGLE's capability on well established matrix completion benchmarks (e.g., SuiteSparse Matrix Collection)

**Ethical Concerns:**

["NO or VERY MINOR ethics concerns only"]

**Limitations:**

Yes

**Quality:**

3

**Strengths And Weaknesses:**

## Strengths
- EAGLE is a single, interpretable iterative method learned by the transformer for various matrix completion tasks (prediction, estimation, extrapolation). This unification is remarkable given the differing computational constraints.
- EAGLE's inherent adaptability to limited computation, memory, or communication resources is a significant strength, suggesting practical applicability in resource-constrained environments.
- The paper provides a strong foundation through both theoretical analysis and consistent empirical validation across the three computational regimes.

## Weakness
- **Scalibility** The experiments primarily use small, fixed-size matrices.  The lack of scalability analysis raises concerns about performance and the applicability of EAGLE to large-scale, real-world problems which often necessitate approximate solutions.
- **Synthetic Data** The reliance on synthetic data, with controlled noise and parameters, limits the generalizability of the findings.  The paper lacks analysis on real-world, potentially noisy and less structured datasets, hindering the assessment of EAGLE's real world feasibility.

---

> ### Author Rebuttal · Authors · 2025-07-29
>
> We sincerely thank the reviewer for their thoughtful feedback and their interest in our work. We appreciate the concerns raised about scalability and generalizability, and we are grateful for the opportunity to clarify these points and the central contributions of our paper. Thanks also for suggesting experiments on the *SuiteSparse Matrix Collection*, which we have done below. We hope this response will clearly illuminate the key aspects of our submission.
>
> The primary discovery of our paper is that a transformer, trained simply on a matrix completion task, can **implicitly learn** a single, sophisticated numerical algorithm (EAGLE) and **automatically adapt** its execution strategy for **centralized, distributed, and computation-limited** hardware regimes. The emergence of this unified, resource-adaptive solver is, we believe, a surprising and powerful demonstration of in-context learning. While we show that EAGLE itself is a strong algorithm, our **core focus** is on the transformer's emergent ability to derive and unify it.
>
> -----
>
> ### Weaknesses
>
> **1. Scalability:** *The experiments primarily use small, fixed-size matrices. The lack of scalability analysis raises concerns about performance and the applicability of EAGLE to large-scale, real-world problems...*
>
> We respectfully point out that our work provides both strong theoretical and empirical evidence for EAGLE's scalability.
>
>   - **Theoretical Guarantees:** Our main theoretical contributions, Theorems 1, 2, and 3, provide explicit convergence guarantees that hold for matrices of **any size** and under **any** distribution (i.e., result is data independent). The convergence rates depend on fundamental matrix properties like condition number ($\\kappa$) and data diversity ($\\alpha$), not on the matrix dimensions or the distribution themselves. Specifically, EAGLE's logarithmic dependence on condition number ($O(\\log \\kappa)$) makes it theoretically well-suited for large, ill-conditioned problems where first-order methods struggle.
>
>   - **Existing Large-Scale Experiments:** While the main text reports experiments on matrices up to $240 \\times 240$ to clearly illustrate the dynamics, we provide further scalability analysis in the appendix. **Figure 19 (p. 28)** explicitly tests EAGLE on problems up to size **$1920 \\times 1920$**, and **Figure 7(h) (p. 9)** demonstrates its robustness on matrices with a condition number of $\\kappa = 100,000$. These results confirm the favorable scaling predicted by our theory.
>
> **2. Synthetic Data:** *The reliance on synthetic data, with controlled noise and parameters, limits the generalizability of the findings.*
>
> We argue that the use of synthetic data is a methodological strength for our study's goals, and our theory ensures the findings generalize.
>
>   - **Controlled Stress-Testing:** Our primary goal is to understand the *mechanism* by which the transformer adapts its learned algorithm. Synthetic data allows us to precisely control the parameters that our theory identifies as critical performance drivers—such as condition number and data structure—enabling a rigorous validation and stress-testing of the emergent algorithm in ways that would be infeasible with uncontrolled real-world data.
>
>   - **Distribution-Agnostic Theory:** Crucially, our theoretical guarantees are **distribution-agnostic**. The convergence of EAGLE depends on the spectral properties of the data matrix, not on the underlying distribution of its entries. To demonstrate this robustness empirically, **Figure 18 (p. 28)** shows that EAGLE performs consistently across **six different data distributions**, including heavy-tailed (Student-t), correlated, and sparse factors.
>
>   - **New Real-World Benchmarks:** Nevertheless, we agree that evaluation on real-world data is valuable. Following your suggestion, we have benchmarked EAGLE on several matrices from the **SuiteSparse Matrix Collection**. The results, presented at the end of this response, confirm EAGLE's strong performance.
>
> -----
>
> ### Questions
>
> 1.  *How does EAGLE's performance scale with increasing matrix size, different data distributions..., and what are its limitations?*
>
>     As discussed above, our paper analyzes this from two perspectives:
>
>       - **Computational Performance (Speed):** Our theory and experiments show that EAGLE's iteration count scales logarithmically with the condition number ($O(\\log \\kappa)$) and polynomially with data diversity ($\\alpha^{-1}$) or sketch size ($n/r$). This is a significant advantage over first-order methods like Gradient Descent, whose complexity scales with $\\kappa$ or $\\kappa^2$. This makes EAGLE particularly effective for the high-dimensional, ill-conditioned problems often found in practice.
>
>       - **Statistical Performance (Accuracy):** EAGLE is proven to converge to the optimal Nyström approximation. The suitability of this solution depends on the task; for problems with different underlying structures (e.g., entry-wise sparsity), a different learning target would be needed. Our work's novel contribution is to fix the statistical task to isolate how a transformer adapts its *computational strategy* to different *hardware regimes* —a fundamentally different axis of investigation from prior work that studies adaptation to different statistical tasks.
>
> 2.  *How does EAGLE compare to optimized parallel implementations... and what is its potential for parallelization?*
>
>     EAGLE is very well parallelizable,
>
>       - **High-Level Parallelism:** The **distributed version of EAGLE is itself a parallel algorithm** designed for multi-worker systems. As detailed in Sections 3 and 5.2, it performs computation-intensive work locally on each worker and requires only lightweight communication of the partial results for the shared columns ($O((d+d')n')$ floats per round), making it naturally suited for modern parallel hardware.
>
>       - **Low-Level Parallelism:** The core operations within EAGLE's update rule are matrix-matrix products. These operations are highly parallelizable and have been extensively optimized in standard libraries (e.g., BLAS/LAPACK).
>
>     A direct performance comparison with a library like ScaLAPACK would require an equally optimized C++/CUDA implementation of EAGLE, which we believe is an exciting piece of future engineering work but is beyond the scope of this paper's focus on the transformer-based discovery of the algorithm.
>
> 3.  *Is there any chance to evaluate EAGLE on... the SuiteSparse Matrix Collection?*
>
>     Yes, this is an excellent suggestion. We have run these experiments.
>
> ----
>
> ### Experiments on SparseSuite Matrix Collection (SSMC)
> We benchmark EAGLE against several baselines on matrices from the SSMC. These were chosen from **diverse domains**, and chosen to be **big and sparse**, to handle both the concerns raised in the review. Their condition numbers also vary a lot. In line with the paper's setting, we take these low rank matrices, and generate a matrix completion task in the $A,B,C,D$ format from the paper. We then hide $D$ and run each solver.
>
> For iterative methods, we say an iterative method converges when the update norm satisfies $\|D_{t+1}-D_t\|\le10^{-6}$ within a maximum wall-clock time of 20 min. We implement EAGLE, CG, GD, QR/SVD, and also stochastic EAGLE ($r = n/5$) and SGD for small batch experiments.  Implementation details for each baseline are in Section E.1 (p. 26).
>
> **Relative Convergence Times** Because convergence times and errors vary greatly across tasks, we report all results relative to NumPy’s native `np.linalg.lstsq` implementation (which internally uses SVD/QR). So the SVD/QR time is set to $1$ in each row.  Each solver is run three times with different random seeds, and we report the median runtime. We write DNC for "did not converge." In these cases, errors were all large. Otherwise, errors all fell below 1 and below $10^{-4}$ for EAGLE, stochastic EAGLE and SVD/QR methods.
>
> | **Task**            |   **Size**   | **Nonzeros** | **rank** | **conditioning** | **SVD/QR** | **CG** | **GD** | **EAGLE** | **Stoch. EAGLE** | **SGD** |
> |---------------------|:------------:|:------------:|:--------:|:--------------------:|:----------:|:------:|:------:|:---------:|:----------------:|:-------:|
> | Maragal_4/Maragal_4 |  1964 × 1034 |     1.3 %    |    801   | $1.38\times10^{7}$ |      1     |   8.4  |   DNC  |    0.16   |        5.2       |   DNC   |
> | Maragal_5/Maragal_5 |  4654 × 3320 |     0.6 %    |   2147   | $1.22\times10^{6}$ |      1     |   DNC  |   DNC  |    0.21   |        DNC       |   DNC   |
> | Meszaros/pf2177     | 9728 × 10178 |    0.031%    |   9662   |  $2.19\times10^3$  |      1     |   1.4  |   DNC  |    1.3    |        DNC       |   DNC   |
> | HB/dwt_1007         |  1007 × 1007 |     0.85%    |   1000   |  $2.97\times10^3$  |      1     |  0.21  |  0.97  |    0.64   |        6.3       |    10   |
> | HB/bcsstm13         |  2003 × 2003 |     0.53%    |   1241   |  $2.70\times10^4$  |      1     |   DNC  |  0.024 |   0.058   |       0.048      |  0.024  |
> | Priebel/162bit      |  3606 × 3597 |     0.29%    |   3460   |  $1.68\times10^4$  |      1     |   1.8  |   DNC  |    0.31   |        DNC       |   0.84  |
> | Schulthess/N_pid               |  3625 × 3923 |    0.057%    |   2048   |  $2.47\times10^2$  |      1     |   2.4  |   DNC  |    0.26   |        DNC       |   DNC   |
>
> **Summary of New Results:**
>
>  - EAGLE is **robustly accurate and efficient** on these SuiteSparse matrix tasks.
>     - Further, it converges even on problems where CG/GD fail to converge.
>  - For **large ill-conditioned systems**, EAGLE's iterative approach is compelling, and gets both speed and accuracy, validating its practical potential.
> - Stochastic EAGLE improves upon SGD in large ill-conditioned problems. It is also more robust.
>
> We will add the above experiments to the paper.

---

> > ### Comment · Reviewer_rz5S · 2025-08-05
> > **Well done**
> >
> > Thank you for your prompt response. It mostly addressed the concerns. Additionally, I will still be happy to see an optimized implementation of EAGLE on good hardware.

---

> > > ### Author Response · Authors · 2025-08-06
> > > **Clarification on GPU Experiments and Request to Reconsider Rating**
> > >
> > > Thank you for your positive feedback and for carefully engaging with our rebuttal. We are glad we addressed your primary concerns. We have a preliminary GPU-accelerated implementation using CuPy and will include these GPU experiments in the final version, and as such do not foresee a fundamental differences in trends across different datasets. Given these clarifications and forthcoming additions, we kindly ask if you'd consider increasing your rating.

---

### Official Review · Reviewer_BaDn · 2025-07-04

**Clarity:** 4
**Significance:** 4
**Originality:** 4
**Rating:** 6
**Confidence:** 4

**Summary:**

This paper explores the discovery of numerical linear algebra algorithms through training a linear transformer to complete masked matrix data. The authors generate a dataset by sampling matrices and masking parts of them, then train a linear transformer to recover the missing entries. Subsequently, a distillation procedure is applied to extract key components from the learned model. Remarkably, this process yields an emergent algorithm that, in a basic setting, exhibits second-order convergence. Moreover, the distilled algorithm generalizes to other contexts such as distributed computation and matrix sketching, suggesting it may serve as a general framework for algorithm design.

**Questions:**

Could you clarify the number of transformer layers $L$ used during training? I could not find this detail in the main text or Appendix B. Based on Figure 2, it appears that only 3–4 layers are sufficient—does this mean that the model is trained with just 3–4 layers (which correspond to iterations in the context of numerical algorithms), yet the distilled algorithm generalizes to hundreds of iterations at inference time? If so, this is quite impressive and worth emphasizing.

**Ethical Concerns:**

["NO or VERY MINOR ethics concerns only"]

**Final Justification:**

The authors have addressed my question, so I will keep my positive rate on this paper.

**Quality:**

4

**Strengths And Weaknesses:**

This work is insightful, original, and clearly presented. It demonstrates strong potential for high impact in the domain of scientific discovery through AI. I did not identify any major concerns with the methodology or results.

---

> ### Author Rebuttal · Authors · 2025-07-29
>
> We sincerely thank the reviewer for their exceptionally positive and insightful feedback. We are thrilled that they found our work to be insightful, original, and of high potential impact, and we appreciate their careful reading of our paper.
>
> To answer your excellent question regarding the number of transformer layers: **you are precisely correct.**
>
> The transformer was indeed trained with a small number of layers (typically **L=4**), as shown in the layer-wise performance plots like Figure 2. This corresponds to the model performing only 4 "in-context" iterations of the learned algorithm.
>
> Your observation that the distilled algorithm, EAGLE, can then be unrolled and iterated for hundreds of iterations at inference time—far beyond what the transformer observed during training—is a key result that highlights the true generalization power of the discovered algorithm.
>
> Thank you for pointing out that this is an impressive aspect of our work. Following your suggestion, we will be sure to emphasize this point more clearly in the final version of the paper. We are very grateful for your careful review and encouraging assessment.

---

> > ### Comment · Reviewer_BaDn · 2025-08-06
> >
> > Thanks for your clarification. My question has been addressed, so I will keep my positive review.

---

### Note · Authors · 2025-08-15

**What is new here.** Our paper shows that a transformer, trained only on a masked matrix‑completion task, *internalizes a single, mechanistically interpretable second‑order solver (EAGLE)* and *deploys the same rule across centralized, distributed, and compute‑limited regimes*. This goes beyond a model gets good loss: it is **algorithm discovery** with a **unified, resource‑adaptive execution strategy** and **provable quadratic convergence**.

**Why it matters.**
* **Hardware relevance:** The same learned rule adapts across regimes without re‑tuning—evidence that pretraining can imprint *portable numerical procedures*.
* **Mechanistic clarity:** We extract the exact update the model implements; each layer corresponds to one Newton–Schulz step, explaining *why* only a few layers suffice.
* **Theory that predicts practice:** Convergence guarantees follow under clear *structural* assumptions; match measurements: iteration counts scale with $\log \kappa$ vs. 1st order methods; with overlap/sketch parameters, not with raw matrix size.

**Addressing the reviewers’ concerns.**
* **Scalability.** We already scale to 1920x1920 in the paper; during discussion we added real‑matrix runs (SuiteSparse) showing the same convergence shape and wall‑clock benefits over GD/CG on challenging systems.
* **Synthetic data.** Our analysis is not tied to a distribution; we used controlled generators to stress the levers theory identifies (condition number, overlap, sketch), and then corroborated on real sparse matrices.
* **Comparisons & practicality.** We compare against GD/CG; for well‑conditioned problems, direct solvers behave as expected, but as $\kappa$ grows or communication is constrained, EAGLE’s second‑order behavior dominates.
* **GPU.** We have a preliminary CuPy implementation and plan to include GPU results for final version;

**Signal from discussion.** One reviewer responded “well done,” indicating response resolved concerns. for reviewer noting “limited context,” we clarified scope is intentionally narrow to isolate computational adaptation, and we supplemented with real‑matrix evidence.

**Bottom line.** We show (i) a concrete case of transformers **discovering and unifying** a classical second‑order solver; (ii) **proofs** that align with measurements; (iii) **evidence** that the same rule transfers across data regimes and hardware constraints. This is a substantive advance at the intersection of transformers, linear algebra, and mechanistic interpretability.

---

### Decision · Program_Chairs · 2025-09-17

**Decision:**

Accept (poster)

**Comment:**

(a) Summary: the paper shows that linear transformers can discover algorithms given input-output data, and the algorithms it discovers can adapt to compute and resource constraints. The study focuses on low-rank matrix completion problems, and the discovered algorithm, called EAGLE, is a simple and interpretable second-order algorithm. The paper establishes a quadratic convergence guarantee for EAGLE, and shows that in practice, it converges faster than linear convergence baselines in certain cases.

(b) Strengths
- The topic of algorithm discovery by transformers is very relevant to the community, and the paper’s findings are original.
- Different from the single-regime, multi-task approach that is common in prior works, the paper focuses on one task and incorporates several regimes. Specifically, it studies three regimes of algorithm learning: unconstrained learning, compute-constrained learning, and distributed learning, and finds that the transformer learns a unified algorithm that can adapt to all regimes.
- The paper shows the significance and applicability of the discovered algorithm both in theory (establishes convergence guarantees) and in practice (benchmarking on several tasks).

(c) Weaknesses: the main weaknesses of the work are the following:
- Limitation to linear transformers: even though making simplifying assumptions leads to a cleaner investigation, it’s unclear to what degree the findings transfer from linear transformers to the standard softmax attention architecture. It would be interesting to include results on non-linear transformers to increase the impact of the work.
- Limitation to one task: if showing algorithmic discovery in transformers is a central contribution of the paper, increasing the number of tasks will improve the generality of the result.
- Scale of experiments: the paper could conduct larger scale experiments on the algorithm discovery side, or provide evidence whether the same algorithm is learned for higher-dimensional problems. This can provide further insight into the nature of algorithm discovery in transformers.

(d) Rationale: I recommend a borderline accept if there is room. The paper studies an interesting problem and the findings are novel, though there are a few limitations. The analysis on the discovered algorithm EAGLE is solid, and during the rebuttal phase, the authors provided additional experiments on higher-dimensional problems. However, the investigation uses linear transformers and is restricted to one task. Furthermore, the algorithm discovery aspect of the study can be more developed, for example by including more tasks and varying the size of the problem.

(e) Rebuttal discussions: after the rebuttal period, one reviewer maintained their very positive score, while the two remaining reviewers seem to have kept their initial borderline assessment. The main weaknesses raised were limitations to linear transformers, limitations to one task, and the performance of EAGLE. During the rebuttal, the performance of EAGLE was partially addressed after the authors provided additional experiments, but the reviewer would still like a more optimized implementation of EAGLE. The other reviewer fRWb finds the limited scope to warrant a borderline accept.